# An FGF15/19-TFEB regulatory loop controls hepatic cholesterol and bile acid homeostasis

Yifeng Wang[1], Sumedha Gunewardena[2], Feng Li[3], David J. Matye[1,4], Cheng Chen[1], Xiaojuan Chao[1], Taeyoon Jung [1], Yuxia Zhang[1], Maciej Czerwiński[5], Hong-Min Ni[1], Wen-Xing Ding [1] & Tiangang Li [4 ✉]

Bile acid synthesis plays a key role in regulating whole body cholesterol homeostasis. Transcriptional factor EB (TFEB) is a nutrient and stress-sensing transcriptional factor that promotes lysosomal biogenesis. Here we report a role of TFEB in regulating hepatic bile acid synthesis. We show that TFEB induces cholesterol 7α-hydroxylase (CYP7A1) in human hepatocytes and mouse livers and prevents hepatic cholesterol accumulation and hypercholesterolemia in Western diet-fed mice. Furthermore, we find that cholesterol-induced lysosomal stress feed-forward activates TFEB via promoting TFEB nuclear translocation, while bile acid-induced fibroblast growth factor 19 (FGF19), acting via mTOR/ERK signaling and TFEB phosphorylation, feedback inhibits TFEB nuclear translocation in hepatocytes. Consistently, blocking intestinal bile acid uptake by an apical sodium-bile acid transporter (ASBT) inhibitor decreases ileal FGF15, enhances hepatic TFEB nuclear localization and improves cholesterol homeostasis in Western diet-fed mice. This study has identified a TFEB-mediated gut-liver signaling axis that regulates hepatic cholesterol and bile acid homeostasis.

[1] Department of Pharmacology, Toxicology and Therapeutics, University of Kansas Medical Center, Kansas City, KS 66160, USA. [2] Department of Molecular and Integrative Physiology, University of Kansas Medical Center, Kansas City, KS 66160, USA. [3] Department of Molecular and Cellular Biology, Baylor College of Medicine, Houston, TX 77030, USA. [4] Harold Hamm Diabetes Center, Department of Physiology, University of Oklahoma Health Sciences Center, Oklahoma City, OK 73104, USA. [5] Sekisui XenoTech LLC, Kansas City, KS 66103, USA. ✉email: tiangang-li@ouhsc.edu

Conversion of cholesterol to bile acids occurs exclusively in hepatocytes[1]. Hepatic bile acid synthesis accounts for about half of the daily cholesterol elimination and has a major impact on whole-body cholesterol homeostasis. Cholesterol 7α-hydroxylase (CYP7A1) catalyzes the first and rate-limiting step in the classic bile acid synthesis pathway. After being released into small intestine, bile acids are efficiently re-absorbed at the distal ileum by the apical sodium-bile acid transporter (ASBT) into enterocytes and returned to liver via portal circulation[1]. Bile acid-activated nuclear receptor farnesoid x receptor (FXR) plays a central role in mediating the bile acid feedback inhibition of bile acid synthesis[2,3]. Intestinal FXR senses elevated bile acids to induce mouse fibroblast growth factor 15 (FGF15) (human ortholog FGF19), which acts as an endocrine hormone to feedback inhibit hepatic *CYP7A1* gene by binding to FGF receptor 4 (FGFR4) on hepatocytes[3,4]. The downstream mechanisms mediating the FGF15/19 signaling repression of the *CYP7A1* gene have not been fully elucidated[3–6].

Bile acid signaling critically regulates lipid, glucose and energy homeostasis and serves as promising therapeutic target for treating metabolic and inflammatory liver diseases[1]. FXR agonist obeticholic acid and engineered FGF19 analog NGM282 have been shown to improve non-alcoholic steatohepatitis (NASH) in clinical trials[7,8]. Studies also suggest that enhanced enterohepatic bile acid circulation and elevated circulating FGF19 may mediate the metabolic improvements after bariatric surgery[9]. Interestingly, emerging evidence shows that altered gut microbiome and the resulting antagonism of intestinal FXR activity protects against metabolic dysfunction[10,11]. Similarly, blocking intestinal bile acid re-uptake improves insulin sensitivity and reduces hepatic steatosis in experimental models and type-2 diabetic patients[12–14]. Intestine-restricted ASBT inhibitors are being tested in clinical trials for treating type-2 diabetes and NASH-associated metabolic disorders[15] (NCT02787304) in addition to their potential clinical application in cholestasis treatment[16]. One of the major consequences of blocking intestinal bile acid recycling is the pronounced and persistent reduction of intestinal FGF15/19 in humans and mice[17,18]. Given the known beneficial effects of bile acids and FGF15/19 signaling, how inhibition of intestinal bile acid re-uptake improves metabolic homeostasis via distinct mechanisms of action requires further investigation.

Transcription factor EB (TFEB) belongs to the microphthalmia family of transcription factors that recognize CLEAR (coordinated lysosomal expression and regulation) DNA elements in the target genes[19]. TFEB has been identified as a nutrient and stress-sensing master regulator of lysosomal biogenesis in various cell types and organ systems[20,21], which has led to a paradigm shift in the understanding of how lysosomal pathways can be dynamically regulated in response to various nutrient and stress signals to maintain cellular homeostasis. Under fed or over-nutrition conditions, nutrient signaling including the mechanistic target of rapamycin (mTOR) and the extracellular signal-regulated kinase (ERK) phosphorylates TFEB on serine residues to cause its cytoplasmic retention[21–23]. Under starvation or lysosomal stress, TFEB is de-phosphorylated and subsequently enters the nucleus to induce a network of genes involved in lysosomal biogenesis and autophagy[20,21]. Studies have demonstrated beneficial roles of TFEB in neurodegenerative diseases and lysosomal storage diseases primarily owing to its stimulation of the cellular clearance pathways[24,25]. Recently, hepatic TFEB overexpression has been shown to protect against non-alcoholic fatty liver disease (NAFLD) and alcoholic liver disease in mice[26,27]. It has been shown that TFEB induces peroxisome proliferator-activated receptor γ co-activator 1α (PGC1α), which activates peroxisome proliferator-activated receptor α (PPARα) to reduce hepatic fat accumulation. Currently, the roles of TFEB in regulating hepatic metabolic pathways are still incompletely understood. Here, we report a gut-liver FGF15/19-TFEB-CYP7A1 regulatory loop that controls cholesterol and bile acid homeostasis and can be modulated by pharmacological inhibition of intestinal bile acid recycling.

## Results

**TFEB induces CYP7A1 in mice and human hepatocytes**. To better understand the impact of TFEB activation on hepatic metabolism, we efficiently knocked down TFEB in mouse livers through adenoviral vector-mediated short-hairpin RNA (shRNA) delivery (Fig. 1a). Gene expression analysis revealed that hepatic TFEB knockdown significantly decreased hepatic CYP7A1 mRNA but not sterol 12α-hydroxylase (CYP8B1) mRNA expression in mice (Fig. 1b). Consistently, hepatic TFEB overexpression induced CYP7A1 mRNA but not CYP8B1 mRNA expression (Fig. 1c, d). As a positive control, TFEB-induced the mRNA of PGC1α (Fig. 1e), a TFEB target gene and strong transcriptional co-activator of the *CYP7A1* gene[28]. Furthermore, TFEB overexpression induced the mRNA of CYP7A1 and the positive control PGC1α but not CYP8B1 in three independent preparations of primary human hepatocytes (Fig. 1f–h), suggesting that TFEB induction of *CYP7A1* gene was hepatocyte autonomous and conserved in humans. Promoter sequence analysis identified several TFEB binding CLEAR elements in human and mouse *CYP7A1* gene promoters within the −1500 bp region (Fig. 2a). Electrophoretic mobility shift assay (EMSA) confirmed that TFEB bound three CLEAR elements in the human *CYP7A1* promoter, two CLEAR elements in the mouse *CYP7A1* promoter and a known CLEAR element in the Nieman-Pick Disease Type-C2 gene promoter (Fig. 2b, c)[20]. Chromatin immunoprecipitation (ChIP) assays performed with human liver and mouse liver tissues confirmed that TFEB binding was significantly enriched in the proximal and distal *CYP7A1* promoter chromatin regions (Fig. 2d, e). These results revealed a role of TFEB in regulating hepatic *CYP7A1* gene expression in mouse livers and human hepatocytes.

**Cholesterol promotes adaptive TFEB nuclear translocation**. Bile acid synthesis is the major cholesterol elimination pathway in the liver. We previously reported that free cholesterol accumulation significantly impaired lysosomal function in hepatocytes[14]. Consistently, free cholesterol loading in HepG2 cells caused free cholesterol accumulation in lysosomes as evidenced by increased number of lysotracker puncta that colocalized with filipin puncta (Fig. 3a). Upon investigating how cholesterol-induced lysosomal stress affects TFEB activation in hepatocytes, we found that nuclear TFEB protein was increased while cytosolic TFEB protein was decreased in response to cholesterol treatment in both HepG2 cells and primary human hepatocytes (Fig. 3b). Increased TFEB nuclear translocation in cholesterol-treated cells were further evidenced by confocal microscopy analysis (Fig. 3c). We recently reported that adding an acyl-CoA cholesterol acyl-transferase (ACAT) inhibitor to cholesterol-treated cells prevented free cholesterol conversion to cholesterol ester, increased cellular free cholesterol and exacerbated lysosomal dysfunction[14] (Supplementary Fig. 1a). Here, we showed that ACAT inhibitor further promoted TFEB nuclear translocation in cholesterol-treated cells (Supplementary Fig. 1b), which supported a link between free cholesterol-induced lysosomal stress and TFEB nuclear translocation. Activation of TFEB by cholesterol loading or amino acid starvation (positive control) correlated with increased lysosome number as indicated by more lysotracker and Lamp1 puncta (Fig. 3a and Supplementary Fig. 1c). Cholesterol loading did not alter mTOR or ERK signaling that is known to

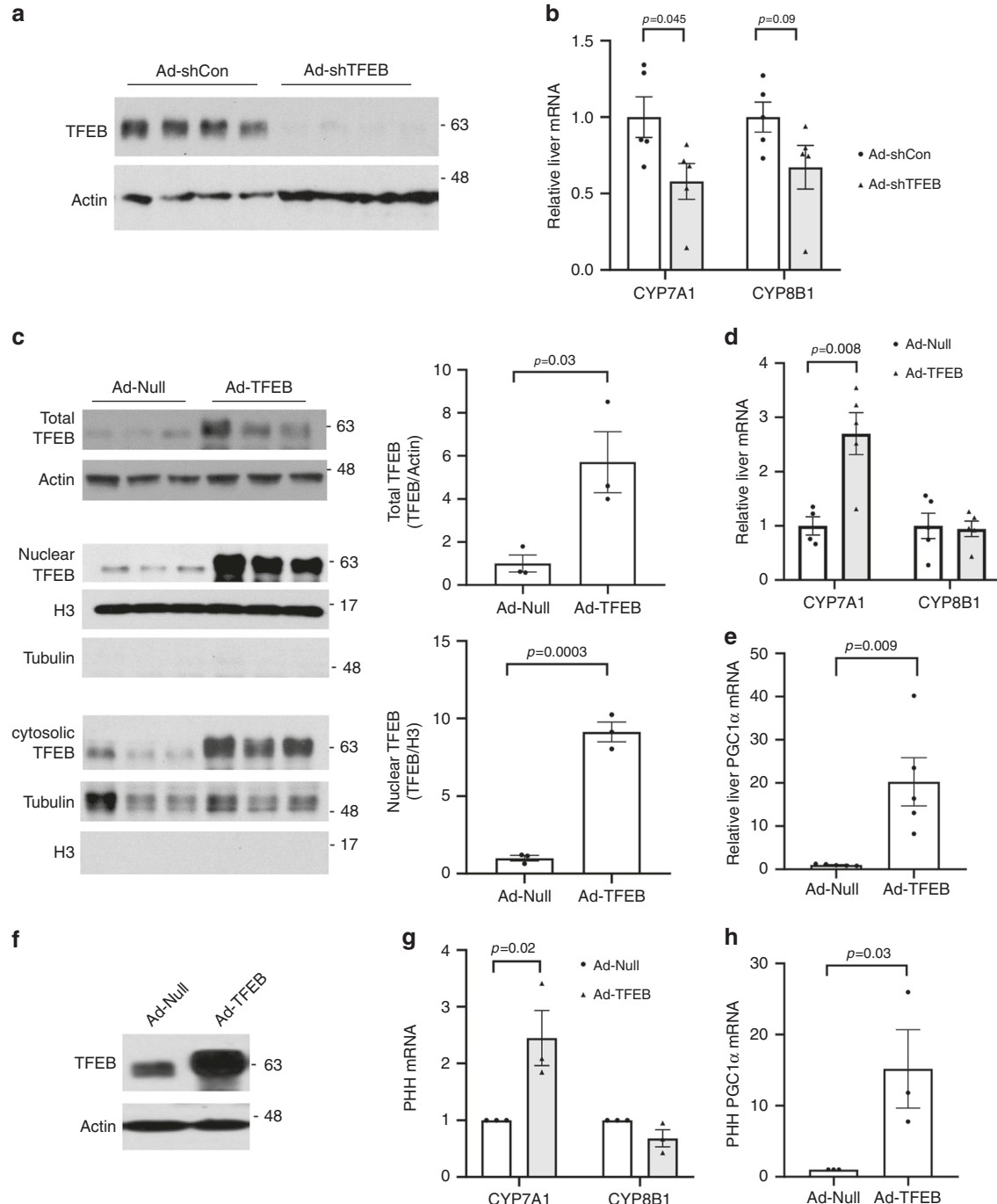

**Fig. 1 TFEB induces hepatic CYP7A1 gene expression. a, b** Male 10-week old C57BL/6J mice were injected Ad-Scramble (Ad-shCon) or Ad-shTFEB at a dose of $1 \times 10^9$ p.f.u. per mouse via tail vein. Mice were fed chow diet for two additional weeks and sacrificed after 6-h fast. ($n = 5$ mice per group). **a** Western blotting of hepatic total TFEB protein. **b** Hepatic mRNA expression. **c–e** Male 10-week old C57BL/6J mice were injected Ad-Null or Ad-TFEB at a dose of $5 \times 10^8$ p.f.u. per mouse via tail vein. Mice were maintained on chow diet for two weeks and sacrificed after 6-h fast. **c** Hepatic total, nuclear and cytosolic TFEB protein. H3: histone 3. TFEB band intensity was normalized to Actin or H3. $n = 3$ mice per group. **d** Relative hepatic CYP7A1 and CYP8B1 mRNA. ($n = 4$ mice per group for Ad-Null; $n = 5$ mice per group for Ad-TFEB). **e** Relative hepatic PGC1a mRNA expression. $n = 5$ mice per group. **f–h** Primary human hepatocytes (PHH) were infected with Ad-Null or Ad-TFEB for 24 h. **f** A representative blot of total TFEB protein of 3 batches of hepatocytes with similar results. **g, h** Average mRNA expression of three independent hepatocyte preparations. All results were expressed as mean ± SEM. Two-sided Student's *t*-test for **b, c, d,** and **e** and one-sided Student's *t*-test for **g, h** was used. Source data for **a–h** are provided as a Source Data file.

regulate TFEB nuclear localization via TFEB phosphorylation (Supplementary Fig. 1d, e). Lastly, we analyzed hepatic TFEB nuclear localization in chronic high-fat/cholesterol Western diet (WD)-fed mice and found that feeding mice WD for 16 weeks, but not 8 weeks, caused significantly increased nuclear TFEB and

decreased cytosolic TFEB in mouse livers (Supplementary Fig. 1f–i). These results suggested that more advanced hepatic lipid accumulation was associated with adaptive TFEB nuclear translocation in mice, which was in agreement with our findings from cholesterol-treated hepatocytes.

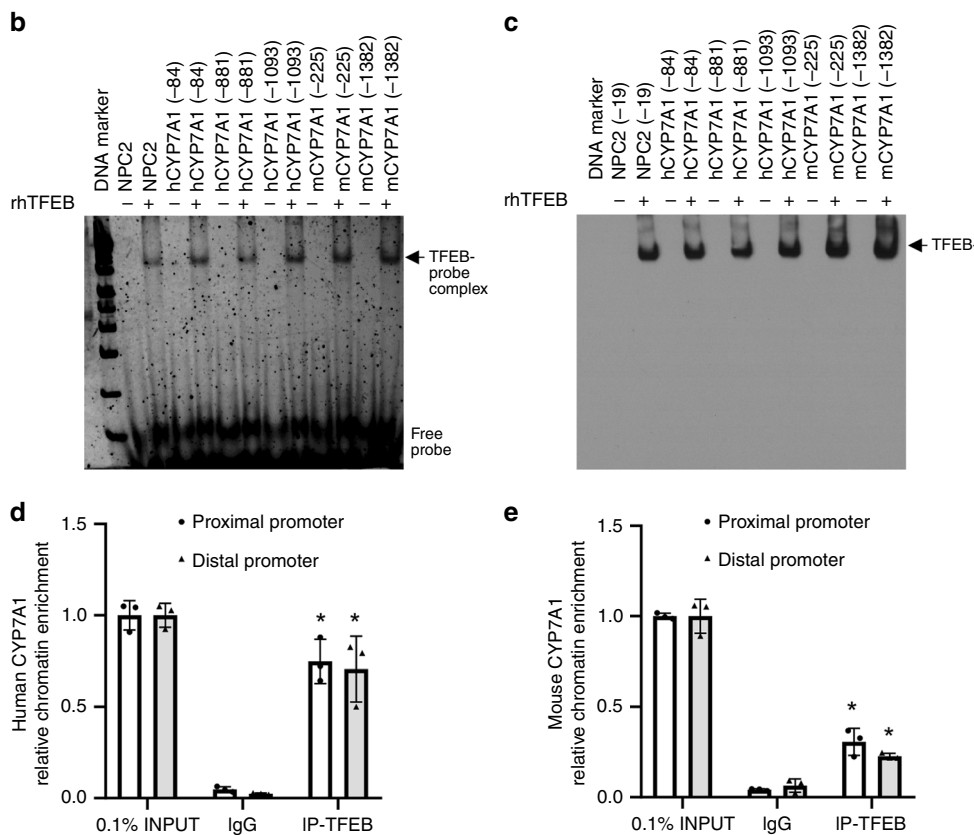

**Fig. 2 TFEB binds human and mouse CYP7A1 promoter. a** The CLEAR sequence location relative to transcriptional start site of *CYP7A1* is indicated in red. **b**, **c** EMSA assay of TFEB binding to CLEAR sequences in human and mouse *CYP7A1* promoter. In-gel imaging of SYBR green signal is shown (**b**). rhTFEB recombinant human TFEB. NPC2 Nieman-Pick Disease Type-C2 (positive control). After SYBR green images were taken, protein in the same polyacrylamide gel was transferred to nitrocellulose membrane and immunoblotted with TFEB antibody (**c**). The arrow indicates the rhTFEB protein with the same mobility as the TFEB-probe complex shown in the left panel. This experiment was repeated once with similar results. **d**, **e** ChIP assay of TFEB binding to human and mouse proximal and distal promoter regions of the *CYP7A1* gene chromatin in human and mouse livers. Pooled normal human livers ($n = 3$) and chow-fed mouse livers ($n = 3$) were used to isolate nuclei for ChIP assay as described in Methods. *, vs. IgG. $p = 0.0006$ for human proximal promoter; $p = 0.003$ for human distal promoter; $p = 0.0035$ for mouse proximal promoter; $p = 0.002$ for mouse distal promoter. Two-sided Student's t-test was used. INPUT (0.1% total) was set as 1. Results were expressed as mean ± SD (technical triplicates). Source data for **b**–**e** are provided as a Source Data file.

**FGF19 inhibits TFEB nuclear translocation in hepatocytes**. Interestingly, we further found that FGF19 treatment decreased both basal nuclear TFEB abundance and cholesterol-stimulated TFEB nuclear localization and increased cytosolic TFEB protein in primary human hepatocytes (Fig. 3d). In HepG2 cells, FGF19 did not affect basal nuclear TFEB abundance but significantly prevented free cholesterol loading or lysosomal inhibitor chloroquine-induced TFEB nuclear translocation (Fig. 3e, f and Supplementary Fig. 1a, b). FGF19 is known to activate several intracellular signaling pathways including ERK and mTOR in hepatocytes (Fig. 4a)[29,30]. It has been reported that ERK and mTOR phosphorylate TFEB at serine residues to cause TFEB cytosolic retention[21–23]. To determine if FGF19 inhibits TFEB nuclear translocation via signaling-dependent mechanism, we next treated HepG2 cells with either the mTOR signaling

inhibitor Torin 1 or the ERK signaling inhibitor U0126 in the presence or absence of FGF19. We found that blocking either mTOR or ERK strongly increased basal TFEB nuclear abundance in HepG2 cells (Fig. 4b and Supplementary Fig. 3a, b), suggesting that mTOR signaling and ERK signaling maintain a strong repression on basal TFEB nuclear translocation. Furthermore, cholesterol-induced TFEB nuclear translocation was repressed by FGF19, but Torin 1 or U0126-induced TFEB nuclear translocation was not affected by FGF19 treatment (Fig. 4b, c). In addition, Torin 1 and U0126 co-treatment did not show additive effect in promoting TFEB nuclear localization in HepG2 cells (Supplementary Fig. 3c, d), suggesting mTOR and ERK may share overlapping downstream mechanisms that control TFEB nuclear localization. In primary human hepatocytes, we found that blocking mTOR by Torin 1

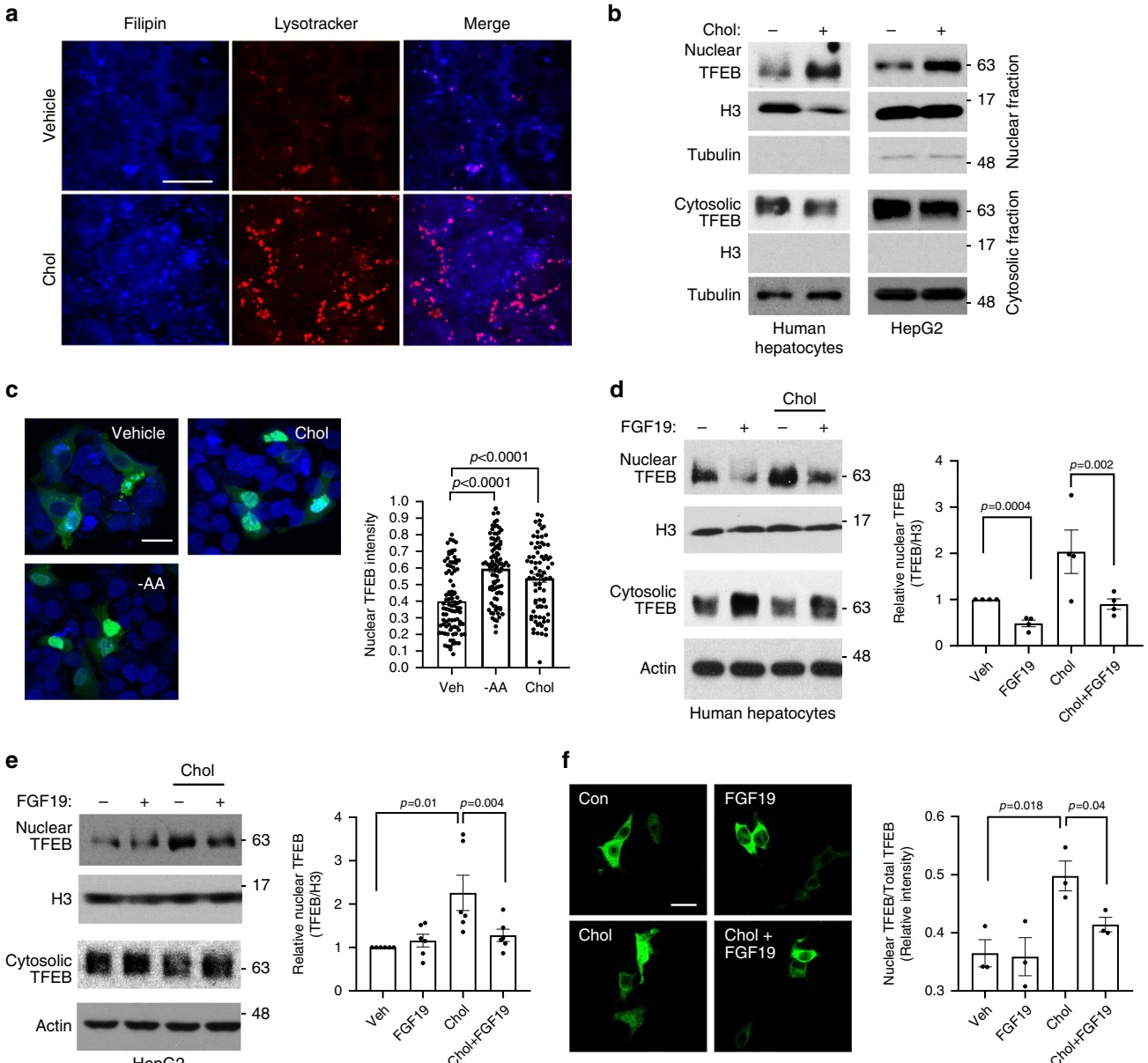

**Fig. 3 FGF19 inhibits lysosomal stress-induced TFEB nuclear translocation. a** HepG2 cells were treated with vehicle (Veh) or 25 μg ml$^{-1}$ cholesterol for 8 h and stained with lysotracker red and filipin. Images were acquired with a confocal microscope. Scale bar: 30 μm. Images are representative of three independent experiments with similar results. **b** Nuclear and cytosolic TFEB protein in human hepatocytes and HepG2 cells treated with 25 μg ml$^{-1}$ cholesterol for 6 h. H3: histone 3. Representative images of at least four independent experiments. **c** HepG2 cells were transfected with TFEB-GFP expression plasmid. After 24 h, cells were treated with 25 μg ml$^{-1}$ cholesterol for 8 h or cultured in amino acid free EBSS culture medium for 3 h. Nuclei were stained with DAPI. Left panel, Confocal microscope was used to acquire images. Scale bar: 20 μm. Right panel, average nuclear/total GFP fluorescent intensity of ~80–100 cells. Results were expressed as mean ± SD. **d** Nuclear and cytosolic TFEB protein. Human hepatocytes were treated with 50 ng ml$^{-1}$ FGF19 for 30 min followed by 25 μg ml$^{-1}$ cholesterol treatment for 6 h. Left panel: A representative blot. Right panel: Average nuclear TFEB protein normalized to histone 3 (H3) in four independent preparations of human hepatocytes ($n = 4$). Control value was arbitrarily set as 1. **e** Nuclear and cytosolic TFEB protein in HepG2 cells. Treatment was the same as in **d**. Left panel: A representative blot. Right panel: Average nuclear TFEB protein normalized to histone 3 (H3) of six independent experiments. Control value was arbitrarily set as 1. **f** TFEB-FLAG expression plasmids were transfected in HepG2 cells. After overnight culture in serum free medium, cells were treated as in **d**. Immunostaining was performed with anti-FLAG antibody. Left panel: representative confocal image (Scale bar: 25 μm). Right panel: Average nuclear/total fluorescent intensity of three independent experiments (Total of 250–360 cells were analyzed per condition). All results in **d**–**f** expressed as mean ± SEM. Two-sided Student's $t$-test was used for **c**–**f**. Source data for **b**–**f** are provided as a Source Data file.

also strongly increased basal TFEB nuclear translocation and completely blocked FGF19 repression of nuclear TFEB localization. In contrast, blocking ERK by U0126 increased basal nuclear TFEB abundance, but did not completely blocked FGF19 repression of nuclear TFEB abundance (Supplementary

Fig. 4a–d), suggesting that in human hepatocytes FGF19 was able to inhibit TFEB nuclear localization via ERK-independent pathways. Despite this cell type-specific difference, our results suggest that FGF19 regulates TFEB nuclear localization via activation of mTOR and/or ERK signaling.

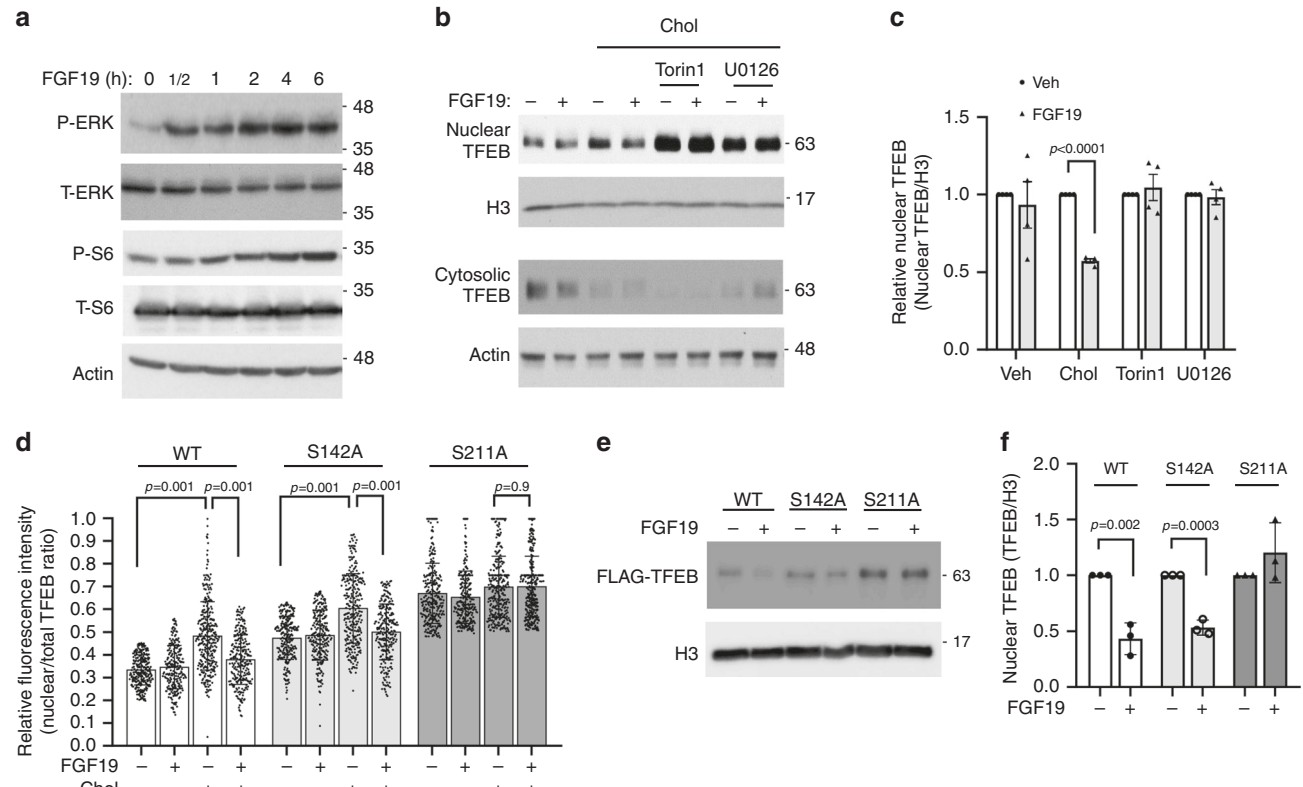

**Fig. 4 FGF19 inhibition of TFEB nuclear translocation requires mTOR/ERK activation and TFEB phosphorylation. a** HepG2 cells were serum starved overnight and treated with 50 ng ml$^{-1}$ FGF19 in time course. Phosphorylated and total ERK and S6 were measured to confirm ERK and mTOR signaling activation. Images are representative of three independent experiments with similar results. **b** Nuclear and cytosolic TFEB abundance in overnight serum starved HepG2 cells treated with 25 μg ml$^{-1}$ cholesterol, 250 nM Torin 1, 10 μM U0126, and 50 ng ml$^{-1}$ FGF19 as indicated for 6 h. H3: histone 3. Images are representative of 4 independent experiments. **c** Same treatment as in **b**. Average nuclear TFEB abundance of 4 independent experiments. **d** WT, S142, and S211 FLAG-tagged TFEB expression plasmids were transfected into HepG2 cells. After overnight culture, cells were treated with 25 μg ml$^{-1}$ cholesterol and/or 50 ng ml$^{-1}$ FGF19 as indicated for 6 h. FLAG-TFEB were detected by immunostaining against FLAG and nuclei were stained with DAPI. Nuclear TFEB/total TFEB of total of 282, 235, 315, 250, 247, 247, 288, 249, 282, 249, 308, and 301 cells per condition (from left to right) from three independent experiments were calculated based on FLAG fluorescent intensity obtained with ImageJ. **e, f** WT, S142A, and S211A FLAG-tagged TFEB expression plasmids were transfected into HepG2 cells. After overnight culture, all cells were treated with 25 μg ml$^{-1}$ cholesterol with/without 50 ng ml$^{-1}$ FGF19 as indicated for 6 h. Nuclear fraction was used to detect FLAG-TFEB and histone 3 (H3). Average nuclear FLAG-TFEB abundance of three independent experiments was shown in **f**. Results in **d** were expressed as mean ± SD. Results in **c** and **f** were expressed as mean ± SEM. Two-sided Student's *t*-test was used for **c** and **f**. Two-way ANOVA and Tukey post hoc were used for **d**. Source data for **a–f** are provided as a Source Data file.

Previous studies showed that mTOR and ERK phosphorylate S142 and/or S211 to inhibit TFEB nuclear translocation in various non-liver cells[21–23], but the roles of these phosphorylation sites in regulating TFEB subcellular localization in liver cells are not clear. To determine the role of S142 and S211 phosphorylation in mediating FGF19 regulation of TFEB nuclear translocation, we expressed FLAG-tagged WT TFEB or S142A and S211A phospho-mutant TFEB in HepG2 cells and studied their subcellular distribution. Under un-treated condition, WT TFEB showed ~30% nuclear localization, TFEB-S142A showed ~45% nuclear localization, while TFEB-S211A showed ~70% nuclear localization (Fig. 4d and Supplementary Fig. 5). Furthermore, FGF19 inhibited the nuclear abundance of both TFEB-WT and TFEB-S142A, but not that of TFEB-S211A (Fig. 4e, f). These results suggest that phosphorylation of S211 plays an important role in TFEB cytosolic retention and is required for FGF19-mediated inhibition of TFEB nuclear translocation. S142 phosphorylation also regulates TFEB cytosolic retention but may not be required for FGF19 to inhibit TFEB nuclear localization.

**FGF19 inhibits hepatic TFEB nuclear localization in mice.** To further determine if FGF19 signaling regulates hepatic TFEB subcellular distribution in vivo, we injected fasted mice recombinant FGF19 for 6 h. We confirmed that FGF19 administration activated mTOR and ERK signaling and inhibited CYP7A1 mRNA expression in mouse livers (Fig. 5a, b). FGF19 administration significantly lowered hepatic nuclear TFEB and increased cytosolic TFEB protein (Fig. 5c, d), which suggests that FGF19 signaling also inhibits hepatic TFEB nuclear translocation in mice in vivo. To determine the role of TFEB in FGF19-mediated CYP7A1 inhibition, we next knocked down liver TFEB and treated these mice with vehicle or FGF19 for 6 h (Fig. 5e). We found that FGF19 strongly inhibited CYP7A1 mRNA by ~80% in the Ad-shCon group (Fig. 5f). Liver TFEB knockdown lowered CYP7A1 mRNA by ~40%, and FGF19 treatment further decreased CYP7A1 mRNA to a level that was comparable to that of Ad-shCon+FGF19 group (Fig. 5f). These results suggest that FGF19 repression of TFEB may partially contribute to FGF19-mediated CYP7A1 inhibition, while FGF19 also inhibits CYP7A1 via other redundant mechanisms independent of TFEB.

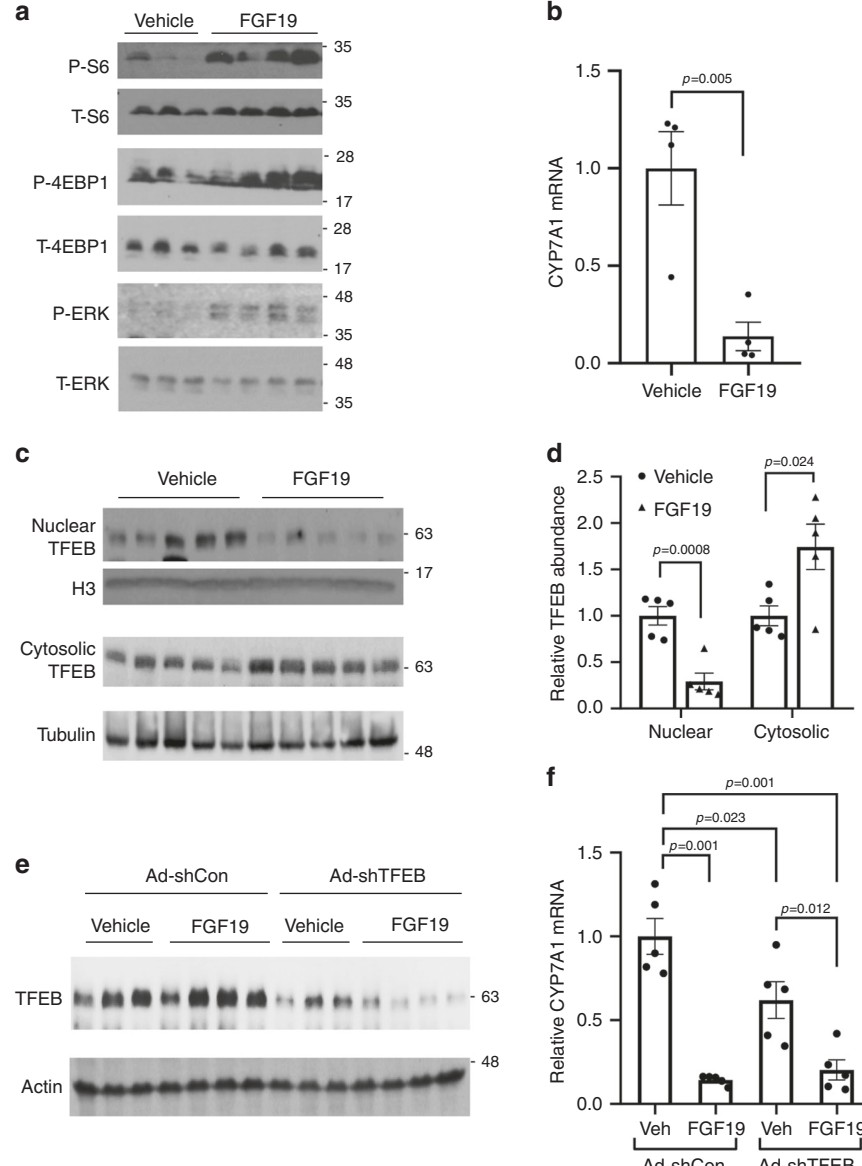

**Fig. 5 FGF19 inhibits hepatic TFEB nuclear localization in mice. a–d** C57BL/6J male mice were fasted overnight and injected with vehicle (sterile PBS) or recombinant FGF19 (1 mg kg$^{-1}$ BW) via tail vein. Mice were sacrificed 6 h later. **a** Hepatic phosphorylated and total S6, 4E-BP1 and ERK. $n = 3$ mice per group for vehicle and $n = 4$ mice per group for FGF15. **b** Liver CYP7A1 mRNA. ($n = 4$ mice per group). **c–d** Nuclear and cytosolic TFEB protein in mouse livers. Relative TFEB abundance was calculated by normalizing nuclear and cytosolic TFEB band intensity to histone 3 (H3) and tubulin, respectively. ($n = 5$ mice/group). **e, f** Male C57BL/6J mice were injected with Ad-scramble (Ad-shCon) or Ad-shTFEB via tail vein. One week later, mice were fasted overnight and injected with vehicle (Veh) or recombinant FGF19 (1 mg kg$^{-1}$ BW) via tail vein and sacrificed 6 h later. **e** Hepatic TFEB protein. **f** Hepatic CYP7A1 mRNA expression. ($n = 5$ mice per group). All results were expressed as mean ± SEM. Two-sided Student's $t$-test was used for **b** and **d**. Two-way ANOVA and Tukey post hoc were used for **f**. Source data for **a–f** are provided as a Source Data file.

**ASBT inhibitor induces hepatic TFEB nuclear localization.** Recently published report has established a protective role of hepatic TFEB activation against NAFLD in mice[26]. We next investigated if pharmacologically targeting the FGF15/19-TFEB axis may be a feasible approach to enhance hepatic TFEB function in vivo. To this end, we treated chow-fed mice with an intestine-restricted ASBT inhibitor GSK2330672 (GSK), which increased fecal bile acid excretion, decreased bile acid pool and markedly lowered ileal FGF15 (Fig. 6a–c). Interestingly, GSK treatment significantly increased hepatic TFEB nuclear abundance without altering total hepatic TFEB mRNA or protein in mice (Fig. 6d, e). Consistent with enhanced hepatic TFEB function, the mRNA expression of hepatic TFEB targets CYP7A1, PGC1α, CPT1, and FGF21 was either significantly elevated or

trended higher in the GSK-treated group (Fig. 6e). These results suggest that blocking intestinal bile acid recycling may be a plausible pharmacological approach to enhance hepatic TFEB function in vivo.

To further investigate the effect of GSK on hepatic TFEB nuclear translocation and metabolic homeostasis in established NAFLD model, we fed mice WD for 10 weeks and treated mice with GSK for 2 weeks. GSK treatment significantly increased hepatic nuclear TFEB abundance in WD-fed mice (Fig. 7a). GSK treatment did not significantly lower body weight but caused marked reduction in hepatic steatosis with reduced hepatic TG and cholesterol accumulation and liver weight/body weight ratio (Fig. 7b–d and Supplementary Fig. 6a–c). To further determine the impact of GSK on hepatic metabolic network, we performed

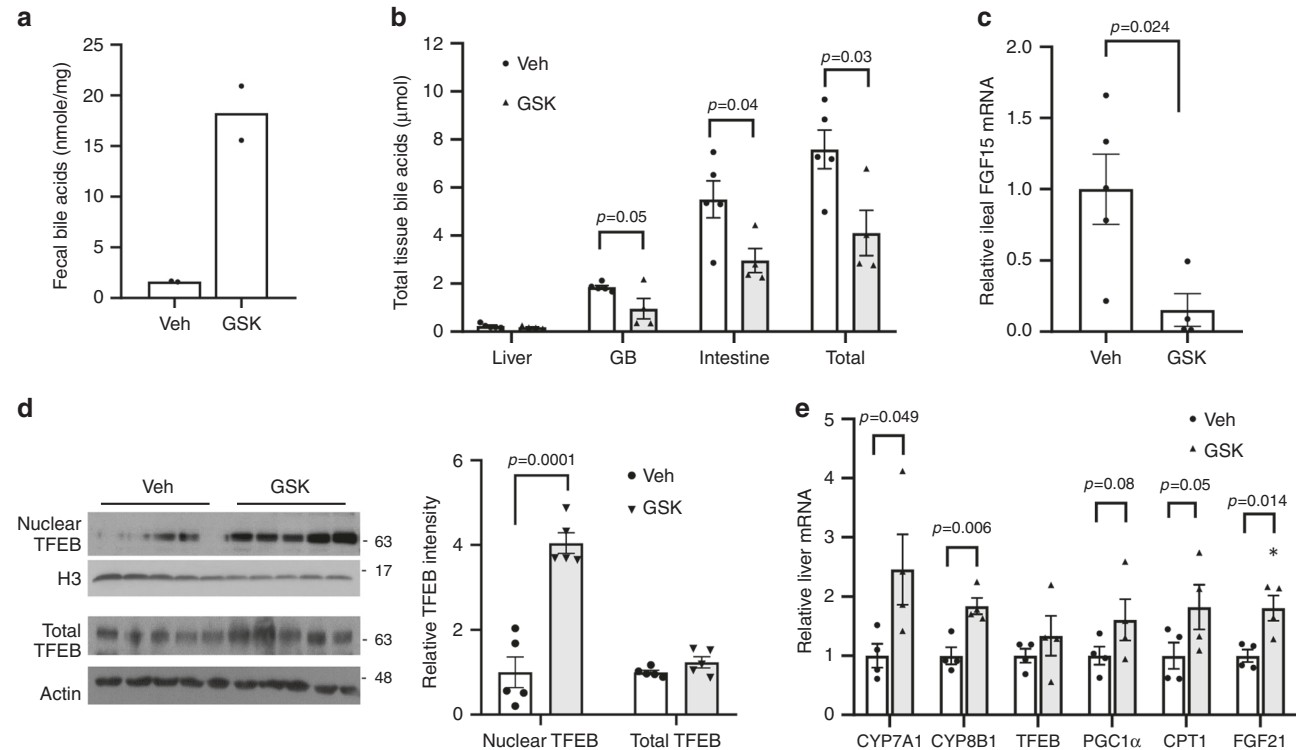

**Fig. 6 ASBT stimulated hepatic TFEB nuclear translocation in mice.** Male C57BL/6J mice on chow diet were treated daily with 2 mg kg$^{-1}$ GSK or vehicle (Veh) via oral gavage for 1 week. **a** Average fecal bile acid content. Feces were collected from two cages per treatment group (2–3 mice per cage). $n = 5$ mice per group for Veh; $n = 4$ mice per group for GSK. Fecal bile acid content was expressed as mean of values from two cages per condition. **b** Tissue bile acid content and bile acid pool. $n = 5$ mice per group for Veh; $n = 4$ mice per group for GSK. **c** Ileal FGF15 mRNA. $n = 5$ mice per group for Veh; $n = 4$ mice per group for GSK. **d** TFEB protein in total liver lysates and nuclear fractions. H3 histone 3. Veh vehicle. Nuclear and total TFEB band intensity was normalized to H3 or Actin, respectively. $n = 5$ mice per group. **e** Relative liver mRNA expression. $n = 4$ mice per group. All results except **a** were expressed as mean ± SEM. Two-sided Student's $t$-test was use for **b**–**e**. Source data for **a**–**e** are provided as a Source Data file.

global metabolomics analysis of 757 metabolites in 8 major metabolic pathways (amino acids, carbohydrates, energy, lipids, nucleotides, vitamins, peptides and xenobiotics). A factor analysis-based filtering followed by hierarchical clustering of the significantly altered metabolites revealed three distinct global expression patterns comprising of 329 metabolites, which demonstrated a strong effect of GSK in reversing the global metabolic changes caused by WD (Fig. 7e and Supplementary Fig. 7a). Top altered pathways in each of the three clusters were listed in Supplementary Fig. 7b–d, which include lipid metabolic pathways. Specific analysis of the lipid metabolic pathways revealed that WD caused marked elevation of hepatic long chain fatty acids, diacylglycerols and acylcarnitines (Fig. 7f–h), which are known to cause hepatic lipotoxicity[31]. GSK treatment significantly lowered several major long chain fatty acids and diacylglycerols (Fig. 7f–g). In addition, decreased hepatic C16:0, C16:1 and C18:1 long chain acylcarnitines in GSK-treated WD-fed mice indicated improved mitochondrial fatty acid oxidation (Fig. 7h)[32]. Consistent with reduced hepatic accumulation of toxic lipid intermediates, the hepatic expression of macrophage chemoattractant protein-1 (MCP-1) and pro-inflammatory cytokines and plasma transaminases were either significantly reduced or trended lower in the WD + GSK group (Supplementary Fig. 8a–c).

**TFEB activation lowers hepatic and plasma cholesterol.** To determine if hepatic TFEB activation induces bile acid synthesis and improves cholesterol homeostasis, we subjected TFEB over-expressing mice and controls to WD feeding for 1 week, a feeding

length that markedly increased hepatic and plasma cholesterol (Fig. 8a, b) but was not long enough to cause appreciable hepatic TG accumulation or hypertriglyceridemia (Supplementary Fig. 9a, b), allowing us to study cholesterol changes independent of TFEB-mediated reduction of hepatic steatosis[26]. We found that hepatic TFEB overexpression significantly reduced WD-induced hepatic and gallbladder cholesterol accumulation and hypercholesterolemia (Fig. 8a–c), which correlated with significantly increased bile acid pool (Fig. 8d). Analysis of bile acid composition revealed that hepatic TFEB overexpression increased tauro-chenodeoxycholic acid (T-CDCA) and tauro-deoxycholic acid (T-DCA) abundance and decreased tauro-muricholic acids (T-MCA) abundance (Supplementary Fig. 10a, b). Altered bile acid composition was not a result of altered hepatic CYP8B1 upon liver TEFB overexpression (Fig. 1d), and the underlying mechanisms remain to be determined. These changes are relatively moderate but may render the bile acid pool more hydrophobic with increased FXR agonists T-CDCA and T-DCA and decreased FXR antagonist T-MCAs[33]. Increased bile acid pool, together with altered bile acid composition, also correlated with increased mRNA of hepatic FXR target genes *SHP* and *OSTβ* (Supplementary Fig. 10c). Gene expression analysis showed that hepatic TFEB overexpression induced the mRNA expression of *CYP7A1*, the known TFEB target gene lysosomal acid lipase (*LIPA*) and PGC1α-PPARα target genes carnitine palmitoyl-transferase 1 (*CPT1*) and Acyl-CoA Oxidase 1 (*ACOX*) (Supplementary Fig. 11). Interestingly, hepatic TFEB overexpression resulted in significantly higher mRNA expression of sterol regulatory element-binding protein 2 (*SREBP2*) and its target genes HMG CoA reductase (*HMGCR*), LDL receptor (*LDLR*), StAR

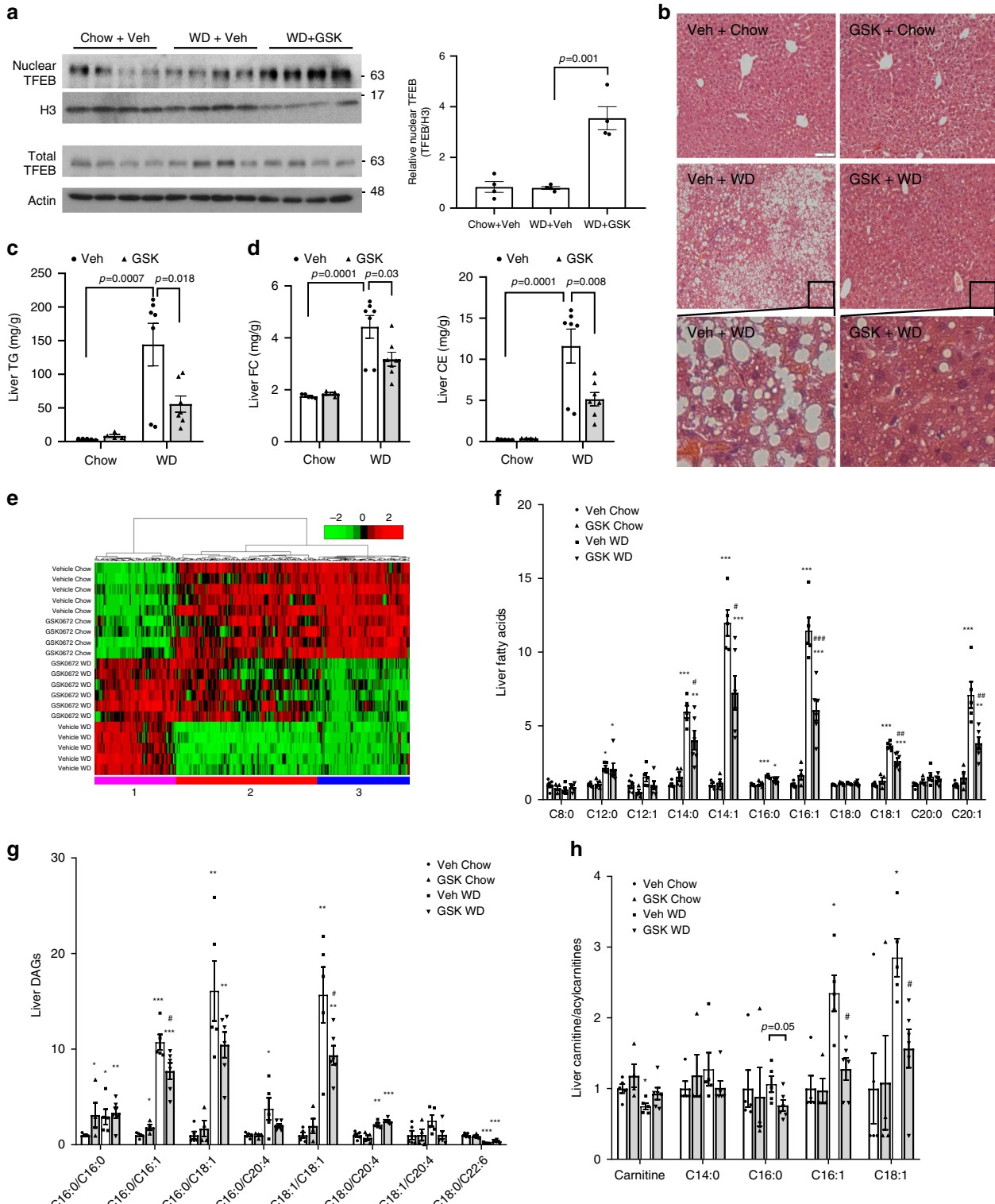

**Fig. 7 ASBT inhibitor improved hepatic steatosis in WD-fed mice.** Male 10-week-old C57BL/6J mice were fed a chow or WD for 10 weeks, and treated with 2 mg kg$^{-1}$ GSK or vehicle (Veh) daily via oral gavage for 2 weeks. **a** TFEB protein in total liver lysates and nuclear fractions. H3 histone 3. Veh vehicle. Nuclear TFEB intensity was normalized to H3 intensity. $n = 4$ mice per group. Two-sided Student's $t$-test was used. **b** Representative hepatic H&E staining of at least four mice per group. Scale bar: 50 μm. **c, d** Hepatic triglyceride (TG), cholesterol ester (CE) and free cholesterol (FC) content. $n = 5$ mice per group for Veh + Chow; $n = 4$ mice per group for GSK + Chow; $n = 7$ mice per group for Veh+WD and GSK + WD. Two-way ANOVA and Tukey post hoc were used. **e** Heatmap of hierarchical clustering analysis. Total of 329 metabolites. Cluster 1 contains 85 metabolites. Cluster 2 contains 147 metabolites. Cluster 3 contains 97 metabolites. **f** Hepatic medium chain and long chain fatty acids. **g** Hepatic diacylglycerols (DAGs). **h** Hepatic carnitine and acylcarnitines. All results were expressed as mean ± SEM. For **f–h**, *$p < 0.05$; **$p < 0.01$; ***$p < 0.001$, vs. Veh + Chow. #$p < 0.05$; ##$p < 0.01$; ###$p < 0.001$, vs. Veh+WD. Two-way ANOVA and Tukey post hoc were used for **f–h**. Source data for **a**, **c**, **d** are provided as a Source Data file. Source data for metabolomics data (**e–h**) are available upon reasonable request.

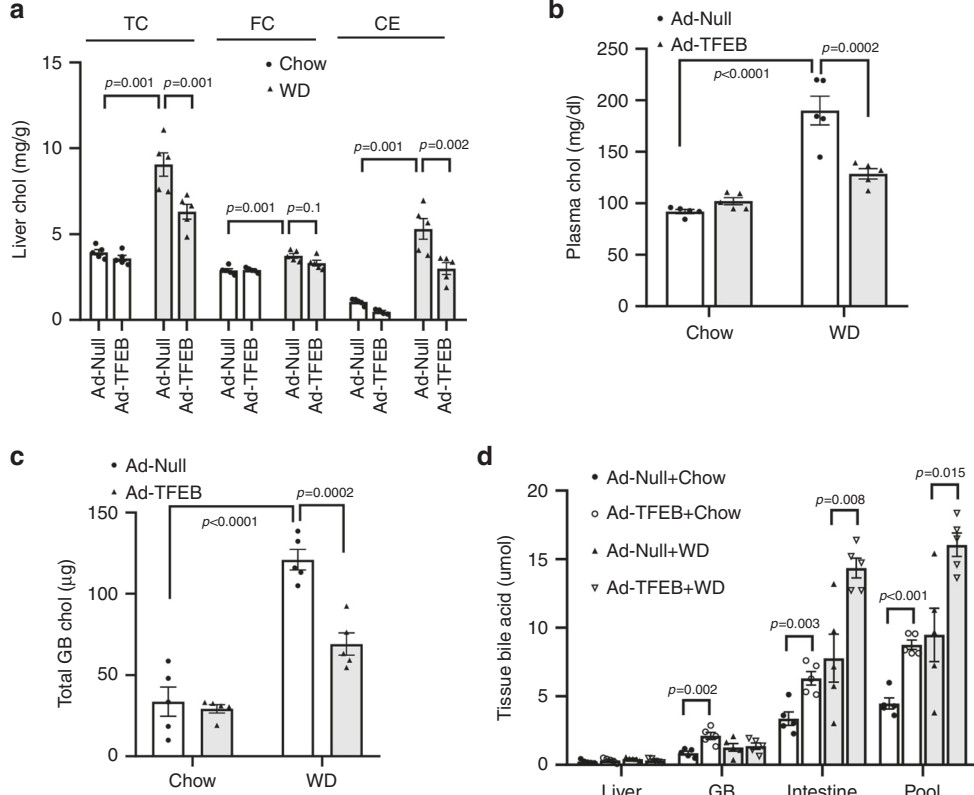

**Fig. 8 Hepatic TFEB induction lowers hepatic and plasma cholesterol in WD-fed mice. a–d** Male 10-week old C57BL/6J mice were injected Ad-Null or Ad-TFEB at a dose of $5 \times 10^8$ p.f.u. per mouse via tail vein. Mice were fed chow diet for one week and then either fed chow diet or challenged with WD for one additional week. ($n = 5$ mice per group). Liver total cholesterol (TC), free cholesterol (FC) and cholesterol ester (CE) (**a**), plasma total cholesterol (**b**), gallbladder (GB) total cholesterol (**c**), and tissue bile acids (**d**) were measured. Bile acid pool is the sum of total bile acids in liver, gallbladder and small intestine with luminal content. All results were expressed as mean ± SEM. Two-way ANOVA and Tukey post hoc were used for **a–c**; Two-sided Student's $t$-test was used for **d**. Source data for **a–d** are provided as a Source Data file.

Related Lipid Transfer Domain Containing 4 (*StARD4*), and squalene epoxidase (*SQLE*) (Supplementary Fig. 11). This was consistent with the previously reported induction of the hepatic SREBP-2 transcriptional network in *CYP7A1* transgenic mice in response to stimulated bile acid synthesis and relative intrahepatic cholesterol reduction[34].

Next, we knocked down hepatic TFEB (Fig. 9a) and subjected hepatic TFEB-deficient mice and controls to 1-week WD challenge, which aimed to determine if hepatic TFEB-deficient mice were more sensitive to WD-induced disturbance of cholesterol homeostasis. We found that hepatic TFEB-deficient mice and controls accumulated comparable levels of cholesterol in the liver and gallbladder on chow diet or after WD challenge (Fig. 9b, c). However, hepatic TFEB-deficiency caused ~35% and ~50% elevation of plasma cholesterol under chow condition and WD-fed condition, respectively (Fig. 9d). When challenged with WD, hepatic TFEB-deficient mice showed significantly increased hepatic VLDL secretion (Fig. 9e), which may explain the absence of intrahepatic cholesterol accumulation and the hypercholesterolemic phenotype in hepatic TFEB-deficient mice. To further determine the relative contribution of hepatic TFEB activation to GSK-mediated cholesterol lowering, we compared the effects of GSK treatment in hepatic TFEB-deficient mice and controls challenged with WD. We found that knockdown of hepatic TFEB to prevent GSK-mediated hepatic TFEB activation did not prevent GSK from lowering hepatic cholesterol content (Fig. 9f). However, GSK failed to lower plasma cholesterol in hepatic TFEB-deficient mice (Fig. 9g). GSK induction of several TFEB-induced genes, including *CYP7A1*, *PGC1α*, and *LIPA* did not appear to be

completely dependent on TFEB (Fig. 9h). Taken together, these results suggest that hepatic TFEB activation partially contributes to improved cholesterol homeostasis in GSK-treated WD-fed mice.

## Discussion

Intestine-derived endocrine hormone FGF15/19 plays a key role in mediating the gut-liver bile acid signaling feedback inhibition of hepatic bile acid synthesis[3,4]. In addition, FGF15/19 signaling has been reported to control cellular metabolism and proliferation in hepatic and extrahepatic tissues, which has important implications in disease pathogenesis and pharmacological treatments[35]. Therefore, better understanding of how FGF15/19 signaling regulates hepatic metabolic pathways is of both basic science and clinical relevance. In this study, we have identified a gut-liver bile acid signaling feedback loop whereby TFEB induces *CYP7A1* to promote bile acid synthesis while bile acid-induced FGF19 in turn inhibits hepatic TFEB activation, which establishes a link between hepatic TFEB and regulation of cholesterol and bile acid homeostasis (Fig. 10). Prior studies have suggested significant signaling redundancy exists in FGF15/19 inhibition of *CYP7A1* and the downstream targets of FGF15/19 signaling remained elusive[4,5,36]. Here, we provide direct evidence that FGF19 treatment activates mTOR and ERK signaling to inhibit TFEB nuclear localization via a phosphorylation dependent manner. Further, hepatic TFEB knockdown is sufficient to cause reduced hepatic *CYP7A1* expression although it does not abolish the maximal repression of *CYP7A1* by FGF19 administration.

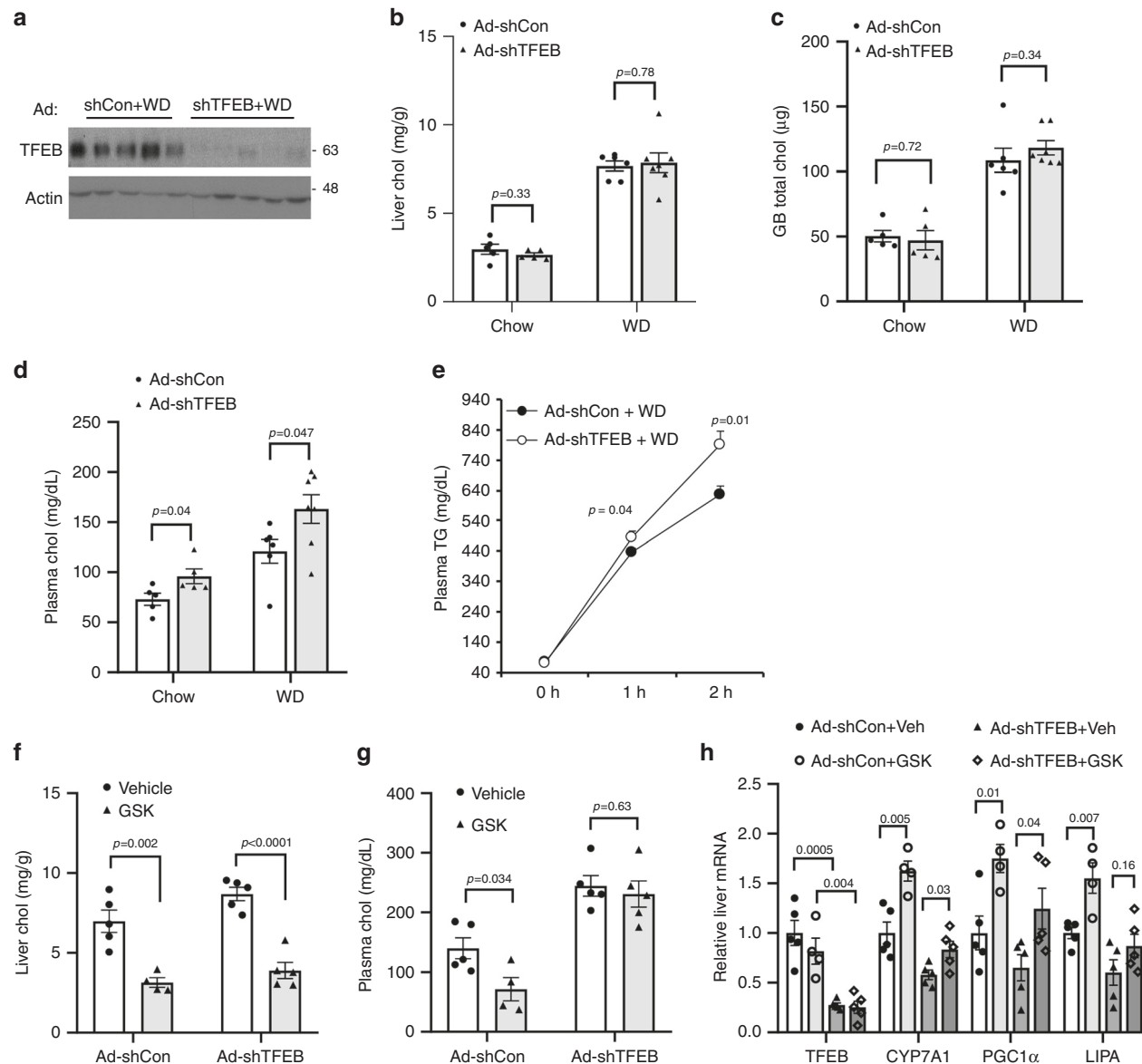

**Fig. 9 Hepatic TFEB knockdown exacerbated hypercholesterolemia in WD-challenged mice. a–d** Male 10-week old C57BL/6J mice were injected Ad-scramble (Ad-shCon) or Ad-shTFEB at a dose of $1 \times 10^9$ p.f.u. per mouse via tail vein. Mice were fed chow diet for one week and then either fed chow or challenged with WD for one additional week. **a**. Liver TFEB protein. $n = 5$ mice per group. **b–d** Liver, gallbladder (GB) and plasma total cholesterol. $n = 5$ mice per group for Ad-shCon + Chow; $n = 5$ mice per group for Ad-shTFEB + Chow; $n = 6$ mice per group for Ad-shCon+WD; $n = 7$ mice per group for Ad-shTFEB+WD. **e** Male 10-week-old C57BL/6J mice were injected Ad-scramble (Ad-shCon) or Ad-shTFEB at a dose of $1 \times 10^9$ p.f.u. per mouse via tail vein. Mice were fed chow diet for one week and then challenged with WD for one additional week. VLDL secretion assay was then performed. ($n = 5$ mice per group). **f–h** Male 10-week old C57BL/6J mice were injected Ad-scramble (Ad-shCon) or Ad-shTFEB at a dose of $1 \times 10^9$ p.f.u. per mouse via tail vein. Mice were either fed chow or challenged with WD for two additional weeks with/without GSK672 (2 mg kg$^{-1}$ day$^{-1}$) treatment. Liver total cholesterol (**f**), plasma total cholesterol (**g**), and liver mRNA (**h**) were measured. $n = 5$ mice per group for Ad-shCon+Veh, Ad-shTFEB+Veh, and Ad-shTFEB+GSK; $n = 4$ mice per group for Ad-shCon + GSK. All results were expressed as mean ± SEM. Two-sided Student's $t$-test was used for **b–h**. Source data for **a–h** are provided as a Source Data file.

These experimental evidences support that TFEB is an inducer of hepatic bile acid synthesis and partially mediates FGF19 feedback inhibition of hepatic *CYP7A1*. Results obtained from primary human hepatocytes suggest that this regulatory mechanism may also be conserved in humans. Our mechanistic studies show that TFEB may induce *CYP7A1* by direct binding to the CLEAR sequence in the *CYP7A1* promoter and by inducing the expression of PGC1α, which is a co-activator of *CYP7A1* gene transcription[28]. In addition to inhibiting TFEB, FGF15/19 has been shown to strongly repress hepatic PGC1α via de-phosphorylation and inactivation of cAMP regulatory element-binding protein (CREB)[37]. From a physiological point of view, bile acids released into small intestine upon food intake induce FGF15/19, which inhibits the rapid postprandial rise of bile acid synthesis in humans and mice[17,38]. Furthermore, many studies have also shown that FGF15/19 acts as an insulin-independent postprandial hormone to stimulate hepatic protein and glycogen synthesis and repress gluconeogenesis and autophagy[30,37,39].

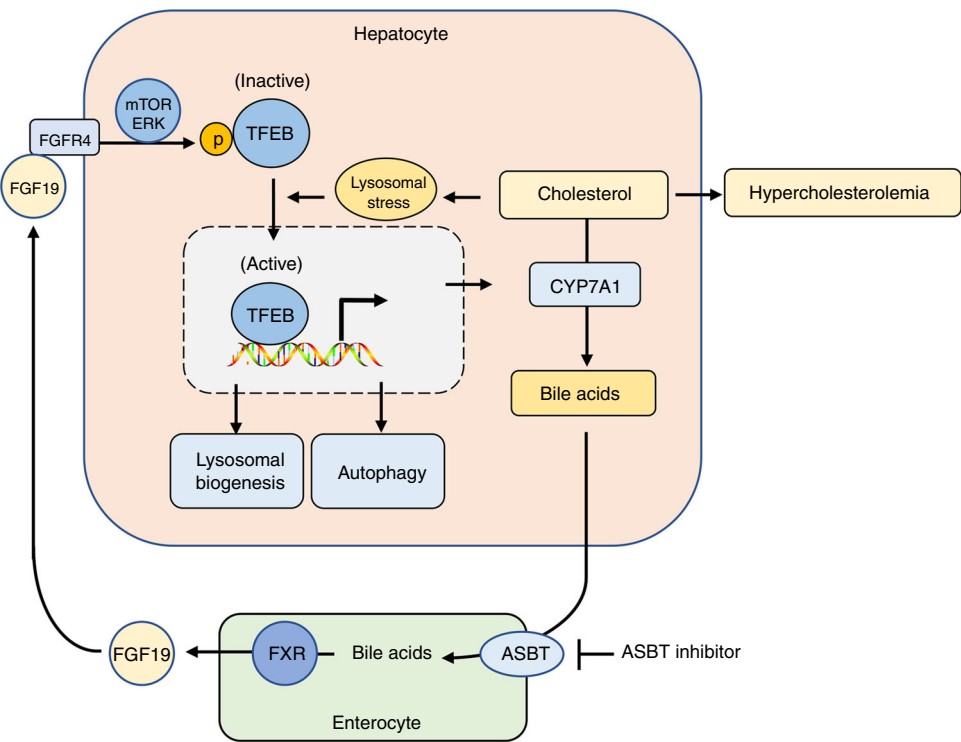

**Fig. 10 TFEB regulation of hepatic cholesterol and bile acid homeostasis.** Hepatocytes process high capability to maintain cholesterol homeostasis via coordinate control of cholesterol synthesis, catabolism and efflux pathways. However, excessive intrahepatic cholesterol accumulation in NASH causes lysosomal stress, mitochondrial dysfunction and oxidative stress. Bile acid synthesis is a major mechanism to prevent intrahepatic cholesterol accumulation and hypercholesterolemia. TFEB is an inducer of lysosomal biogenesis, autophagy and mitochondrial function in response to nutrient deprivation and lysosomal stress. Findings from this study show that excessive intracellular cholesterol accumulation causes lysosomal stress and subsequent TFEB nuclear translocation. TFEB activation induces CYP7A1 to promote bile acid synthesis, which promotes cholesterol catabolism and elimination. In addition, bile acids activate FXR to induce intestinal FGF15/19 to feedback inhibit TFEB by causing TFEB phosphorylation and cytosolic retention. Pharmacological inhibition of ASBT decreases intestinal bile acid-induction of FGF15/19 to potentiate liver TFEB activation, which contributes to the protection against hepatocellular organelle stress and metabolic disorders.

Therefore, FGF15/19 inhibition of TFEB function is consistent with TFEB being a catabolic pathway activator in response to fasting signals[24,40,41].

Studies so far have attributed the cyto-protective effects of TFEB largely to activation of the autophagy-lysosome clearance pathway in various cell types and organ systems, and activation of TFEB has been thought to hold promise for treating lysosomal storage disorders and neurodegenerative diseases[20–23]. More recently, hepatic TFEB overexpression has been shown to induce the PGC1α/PPARα pathway to promote fatty acid oxidation and prevent chronic high-fat diet-induced hepatic steatosis in mice[26]. In addition to fat accumulation, strong evidence suggests that disrupted cholesterol homeostasis and intrahepatic free cholesterol accumulation contribute to organelle dysfunction and lipotoxic liver injury in NASH[42,43]. Human patients with *CYP7A1* mutations developed hypercholesterolemia and premature atherosclerosis[44]. In this study, we show that hepatic TFEB deficiency reduces hepatic *CYP7A1* expression and sensitizes mice to hypercholesterolemia upon WD challenge. On the other hand, hepatic TFEB overexpression prevents hepatic and plasma cholesterol elevation independent of hepatic steatosis, which links TFEB-induction of bile acid synthesis to maintenance of cholesterol homeostasis. We also show that TFEB overexpression in mice results in hepatic *CYP7A1* induction and a ~2-fold expansion of bile acid pool, which is comparable to that of *CYP7A1* transgenic mice[45]. Previous studies have shown that transgenic *CYP7A1* expression in mice has a profound impact on hepatic cholesterol metabolic pathways and

completely prevents diet-induced hepatic steatosis and hypercholesterolemia[45]. In addition to induction of bile acid synthesis, TFEB activation of lysosomal function may add additional benefits in maintaining cholesterol homeostasis[24]. This is because hepatocytes acquire large amounts of cholesterol via receptor-mediated uptake of circulating lipoproteins. Lipoproteins are delivered via the endocytic pathway to lysosomes where lysosomal acid lipase (LIPA) hydrolyzes cholesterol ester to free cholesterol for subsequent intracellular redistribution and bile acid synthesis[46,47]. *LIPA* is a known TFEB target[20]. Decreased hepatic cholesterol ester hydrolysis has been shown to hinder bile acid synthesis by reducing substrate (cholesterol) flux[48,49]. Despite the high adaptability of hepatocytes to attenuate cholesterol-induced organelle stress by coordinately regulating cholesterol esterification, efflux and catabolic pathways, impaired lysosomal function, reduced lysosomal acid lipase activity and cholesterol lipotoxicity have been reported in chronic human NASH livers, which was suggested to contribute to NASH pathogenesis[42,50,51]. In WD-fed mouse models of NAFLD, TFEB nuclear translocation was increased after 16 weeks but not 8 weeks of WD feeding. This suggests that TFEB may be activated in response to lysosomal stress only at a more advanced disease stage during the development of NAFLD. Although under this chronic pathological condition the causes of lysosomal stress are likely complex and multi-factorial, findings from cholesterol-laden hepatocyte models suggest that severe hepatic cholesterol accumulation likely plays an important role in this TFEB adaptive response in NAFLD. At the cellular levels, cholesterol loading did not affect

mTOR and ERK signaling that plays important roles in regulating TFEB nuclear translocation. We have previously shown that cholesterol loading in hepatocytes cause marked lysosomal stress and autophagy impairment[14]. These findings collectively suggest that cholesterol-induced lysosomal stress may promote TFEB nuclear localization independent of its crosstalk with nutrient sensing signaling cascades. Further studies are still needed to dissect the mechanisms of cholesterol regulation of TFEB function in hepatocytes.

Given the numerous beneficial effects of TFEB in regulating cellular homeostasis, pharmacological approaches that enhance TFEB nuclear translocation have also been investigated as potential therapeutics to treat metabolic diseases[52,53]. Here, we show that inhibiting intestinal bile acid re-uptake increases hepatic TFEB nuclear localization and improves hepatic metabolic homeostasis. These findings are in agreement with the gut-liver FGF15/19 inhibition of hepatic TFEB function and place TFEB as a downstream effector of ASBT inhibition. By using hepatic TFEB knockdown approach, we further show that preventing hepatic TFEB activation abolishes the beneficial effects of GSK against diet-induced hypercholesterolemia, suggesting that hepatic TFEB activation is indeed required for GSK-mediated overall cholesterol homeostasis in response to high dietary cholesterol challenge. In addition to promoting bile acid synthesis, TFEB has previously been shown to promote mitochondrial function[54] and lysosomal biogenesis[20,21]. These mechanisms may collectively contribute to the protective effect of enhancing hepatic TFEB function against chronic high-fat diet-induced metabolic disorders[26]. In various extrahepatic tissues, TFEB activation has also been shown to improve cellular homeostasis and prevent organelle dysfunction[24]. Whether attenuation of FGF15/19 signaling through ASBT inhibition modulates TFEB function in extrahepatic tissues will depend on tissue-specific responsiveness to FGF15/19 signaling, which still requires further investigation. It should be noted that GSK treatment still lowers intrahepatic cholesterol content in hepatic TFEB-deficient mice, which can be attributed to TFEB-independent effects of GSK treatment. Metabolomics analysis has revealed many altered metabolic pathways in GSK-treated mice, some of which are likely results of improved overall metabolic homeostasis while other may be causative. Further, ASBT inhibitors have been shown in both humans and animal models to improve insulin sensitivity in diabetes, which may at least in part be attributed to the induction of gut glucagon-like peptide 1[12,55,56]. Findings from this study and others suggest that blocking intestine bile acid re-uptake impacts various metabolic pathways in hepatic and extrahepatic tissues to bring about therapeutic benefits in various metabolic and inflammatory diseases. Currently, ASBT inhibitors have shown therapeutic potential for treating cholestasis, hyperlipidemia, type-II diabetes and NASH in preclinical and clinical studies[12,13,57], but the mechanisms of action are still incompletely understood. Future studies defining the mechanisms underlying the beneficial effects of disrupting intestinal bile acid re-uptake will be of great basic research and translational significance.

In summary, this study has identified TFEB as an inducer of hepatic bile acid synthesis, which mediates the effect of hepatic TFEB in regulating cholesterol homeostasis. In addition, we report a gut-liver FGF15/19 signaling axis that controls hepatic TFEB function, which can be pharmacologically targeted at the level of intestinal bile acid transport to improve metabolic homeostasis.

## Methods

**Reagents**. Anti-TFEB antibody (A303-673A, 1:1000 dilution) was purchased from Bethyl Laboratoryies, Inc (Montgomery, TX). Actin antibody (ab3280, 1:10000 dilution) and α-tubulin antibody (ab7291, 1:2000 dilution) were purchased from Abcam (Cambridge, MA). GSK2330672 was purchased from MedChem Express (Monmouth Junction, NJ). Water soluble cholesterol (complexed to methyl-β-cyclodextrin), filipin, tyloxapol, chloroquine, ACAT inhibitor SANDOZ 58-035, and FLAG antibody (F1804, 1:2000 dilution) were purchased from Sigma-Aldrich (St. Louis, MO). Lysotracker red was purchased from Thermo-Fisher Scientific (Waltham, MA). Antibodies against p-ERK1/2 (T202/Y204, #4370, 1:2000 dilution), total ERK1/2 (#9102, 1:2000 dilution), phospho-S6 ribosomal protein (#2215, 1:2000 dilution), total S6 ribosomal protein (#2217, 1:2000 dilution), p-4E-BP1 (Thr37/46, #2855, 1:2000 dilution), total 4E-BP1 (#9644, 1:2000 dilution), and histone 3 (#9717, 1:2000 dilution), mTOR inhibitor Torin 1, and ERK1/2 inhibitor U0126 were purchased from Cell Signaling Technology (Danvers, MA). Lysosomal-associated membrane protein-1 (Lamp1, 1D4B, 1:500 dilution) was purchased from the Developmental Studies Hybridoma Bank (Iowa City, IA). Recombinant FGF19 was purchased from RnD Systems (Minneapolis, MN). Aspartate aminotransferase (AST) and alanine aminotransferase (ALT) assay kits, total cholesterol assay kit and TG assay kit were purchased from Pointe Scientific (Canton. MI). Free cholesterol assay kit was purchased from Wako Diagnostics (Richmond, VA). Bile acid assay kit was purchased from Diazyme Laboratories (Poway, CA). FLAG-tagged WT, S142A and S211A expression plasmids and TFEB-GFP were generous gifts from Dr. Andrea Ballabio (Baylor College of Medicine, Houston, Texas).

**Mice and treatments**. WT male C57BL/6J mice were purchased from the Jackson Lab (Bar Harbor, ME). The standard chow diet was PicoLab Rodent Diet 20 (LabDiet, St. Louis, MO) containing 13 kcal% fat calories. Western diet (WD) contains 42 kcal% fat calories and 0.2% cholesterol (TD.88137, Envigo, Denver, CO). GSK2330672 (GSK) was prepared in 0.5% methylcellulose and administered via gavage in 200 μl volume in a daily dose of 2 mg kg$^{-1}$ according to published data[12]. Mice were housed in micro-isolator cage with corn cob bedding under 7 a.m.–6 p.m. light cycle and 6 p.m.–7 a.m. dark cycle. Equal volume of vehicle was administered to control mice. Recombinant human FGF19 was dissolved in sterile phosphate-buffered saline (PBS) and injected via tail vein at 1 mg kg$^{-1}$ BW. Adenovirus was injected via tail vein. Mice were fasted for 6 h from 9 a.m. to 3 p.m. before sacrifice except otherwise noted. All animal protocols were approved by the Institutional Animal Care and Use Committee at the University of Kansas Medical Center.

**Triglyceride, cholesterol, and bile acid analysis**. Lipids were extracted in a mixture of chloroform: methanol (2:1; v-v), dried under nitrogen, and resuspended in isopropanol containing 1% triton X-100. Total cholesterol, free cholesterol and TG were measured with assay kits following the manufacturer's instruction. Cholesterol ester was calculated by subtracting free cholesterol from total cholesterol. Bile acids were extracted from liver, whole-gallbladder bile, whole-small intestine with content, and dried fecal samples in 90% ethanol[58]. Total bile acid concentration was measured by assay kit according to the manufacturer's instruction. Bile acid pool was calculated as the sum of total bile acid amount in liver, gallbladder and small intestine. Bile acid composition in gallbladder bile was measured by liquid chromatography–mass spectrometry method[59].

**Metabolomics and bioinformatics analysis**. Metabolomics was performed by Metabolon Inc (Durham, NC). Mouse liver samples were prepared using the automated MicroLab STAR® system from Hamilton Company. Recovery standards were added proportional to tissue weight prior to sample extraction for data normalization. The liver extracts were analyzed on a Waters ACQUITY UPLC and a Thermo Scientific Q-Exactive high-resolution/accurate mass spectrometer interfaced with heated electrospray ionization (HESI-II) source and Orbitrap mass analyzer. Raw data was extracted, peaks were identified and processed using Metabolon's reference library and software. Peaks were quantified using area-under-the-curve and results were expressed as relative fold changes to chow-fed control group. The raw read counts for each biochemical was rescaled to set the median equal to 1 and log transformed for statistical analysis. The relationship between the observed metabolite expression and the two factors, diet and treatment, was modeled using a two-way analysis of variance. The geometric average of metabolite expression of the different cohorts was compared to each other. The statistical significance of these comparisons was calculated using orthogonal contrasts. The resulting $p$-values were adjusted for multiple hypothesis testing using the Benjamini-Hochberg method[60], giving a false discovery rate (FDR) for each metabolite. Metabolites with an absolute fold-change greater than or equal to 1.5 and an FDR less than or equal to 0.1 were considered significantly different between the compared cohorts. A global metabolite expression pattern, i.e., metabolites that express similarly across different cohorts, was obtained by performing a factor analysis-based filtering as previously described by the authors[61]. These metabolites were clustered using a hierarchical clustering algorithm that utilized a Euclidian distance matrix (pairwise distance measure between metabolites across cohorts) and a Ward linkage function. A principal component analysis (PCA) of the metabolite expression data was performed to transform this multi-dimensional data to a two-dimensional plane for visualization. GraphPad Prism 6, R v3.5.1, and MATLAB R2017a software was used for the above analysis.

**Hepatic VLDL secretion assay**. Male C57BL/6J mice at the age of 10 weeks were i.v. injected with Ad-scramble or Ad-shTFEB at a dose of $1 \times 10^9$ p.f.u. per mouse. One week post injection, mice were challenged with WD for 1 week. Mice were fasted for 6 h and VLDL secretion assay was performed as previously described[62]. Briefly, mice were i.v. injected with 300 mg kg$^{-1}$ taloxpol (diluted in PBS in 200 µl injection volume). Blood was collected at indicated time and TG concentration was measured with TG assay kit according to the manufacturer's instruction.

**Nuclear and cytosolic fractionation**. Liver homogenates or cell lysates were prepared in modified RIPA buffer containing 1% NP-40 with a dounce homogenizer. After incubation on ice for 30 min, the lysates were passed through a 28-gauge insulin needle a few times and loaded on top of a sucrose gradient. After centrifugation, cytosolic fraction was removed and nuclei pellet was resuspended in 1X RIPA buffer. The cytosolic fraction and the nuclear fraction were mixed with equal volume of 2X laemmli buffer, sonicated briefly, incubated at 95 °C for 5 min, and used for sodium dodecyl sulfate–polyacrylamide gel electrophoresis analysis.

**Chromatin immunoprecipitation assay**. Pooled normal human livers ($n = 3$) (provided by the KU Liver Center) and chow-fed mouse livers ($n = 3$) were used to isolate nuclei as described above. ChIP assays were performed with anti-TFEB antibody (A303-673A, Bethyl Laboratoryies, Inc. Montgomery, TX. 10 µg ml$^{-1}$ final concentration), Normal rabbit IgG (#2729S, Cell Signaling Technology, Danvers, MA. 10 µg ml$^{-1}$ final concentration) as negative control, Dynabeads™ Protein G magnetic beads (Thermo-Fisher Scientific, Waltham, MA), and a ChIP assay kit (MilliporeSigma, Burlington, MA) following the manufacturer's instruction[34]. The sequence of ChIP real-time PCR primers are: mouse CYP7A1 proximal promoter (−219/−163): Forward: ACCTTCGGCTTATCGACTATTGC; Reverse: TATCTGGCCTTGAACTAAGTCCA TCT. Mouse CYP7A1 distal promoter (−1483/−1400): Forward: GAGGGTCGCTTG GCTTTAAA; Reverse: TCTGAGGTAAGGAGAAAGGAAAACAT. Human CYP7A1 proximal promoter (−180/−111): Forward: GGTCTCTGATTGCTTTGGAACC; Reverse: AAAAGTGGTAGTAACTGGCCTTGAA. Human CYP7A1 distal promoter (−1177/−1042): Forward: ACTCACCAAGTTGATCCTTGAC, Reverse: TGGGCTCT CTGAAATTGTGAC. Amplicon position was relative to the transcriptional start site.

**Cell culture**. HepG2 cells were purchased from the American Type Culture Collection (Manassas, VA). Cells were maintained in Dulbecco's modified Eagle medium supplemented with 10% fetal bovine serum and 1% penicillin/streptomycin. When HepG2 cells were ~90% confluent, they were serum starved overnight before treatments were initiated as indicated. Primary human hepatocytes were obtained from the Cell Isolation Core at KUMC. Primary hepatocytes were seeded in collagen-coated plates and cultured in Williams E medium supplemented with 1% penicillin/streptomycin without serum. Treatments in primary hepatocytes were initiated within 24 h after the cells were plated.

**Recombinant adenovirus**. Ad-Null, Ad-scramble, Ad-TFEB, and Ad-shTFEB were purchased from Vector Biolabs (Philadelphia, PA). Adenovirus was purified from HEK293A cells by CsCl centrifugation subjected to desalting with GE healthcare PD-10 Sephadex G-25 desalting columns (Thermo Fisher Scientific, Grand Island, NY). Adenovirus titer was determined with an Adeno-X rapid titer kit from Clontech (Mountain View, CA). Mice were injected 0.5–1 × 10$^9$ p.f.u. per mouse adenovirus via tail vein.

**Confocal microscopy**. GFP-tagged or FLAG-tagged TFEB expression plasmids were transfected into HepG2 cells using Lipofectamine 3000 reagent (Thermo Fisher Scientific, Grand Island, NY). Treatments were initiated 24 h after transfection. HepG2 cells were then fixed in 4% paraformaldehyde and permeabilized with 0.1% tween-20 and 0.3 M glycine. Primary anti-FLAG antibody (1:2000 dilution) and Anti-LAMP1 antibody (1:2000 dilution) and Alexa Fluor 488-conjugated secondary antibody (A32723, Thermo-Fisher Scientific, 1:2000 dilution) were used for immunofluorescent staining. Images were acquired with either a Leica DM 5500 confocal microscope. Relative fluorescent intensity in nucleus and cytosol were measured with ImageJ software to determine the nuclear TFEB abundance.

**Electrophoretic mobility shift assay (EMSA)**. An Electrophoretic Mobility-Shift Assay (EMSA) Kit with SYBR green detection (Thermo Fisher Scientific, Grand Island, NY) was used following the manufacturer's instruction as previously described[63]. Recombinant human TFEB protein (H00007942-P01) was purchased from Abnova (Walnut, CA). DNA probes were chemically synthesized. In vitro binding reaction contains 5 pmol annealed DNA probe and 200 ng recombinant TFEB protein. The sequence of DNA probes used in EMSA was shown in Fig. 2a. Images were acquired with a LI-COR Odyssey Imaging System.

**Western blotting**. Liver lysates were prepared by placing liver homogenates in RIPA buffer containing 1% SDS and protease inhibitors on ice for 1 h followed by brief sonication. After centrifugation, supernatant was used for SDS-PAGE and immunoblotting. ImageJ software was used to quantify band intensity. TFEB band intensity was normalized to loading control actin, tubulin or histone 3 band intensity and shown as relative band intensity.

**Real-time PCR**. Total RNA was purified by Trizol (Sigma-Aldrich, St. Louis, MO). Reverse transcription was performed with Oligo dT primer and SuperScript III reverse transcriptase (Thermo Fisher Scientific, Grand Island, NY). Real-time PCR was performed with iQ SYBR Green Supermix (Bio-rad, Hercules, CA). Relative mRNA expression was calculated using the comparative CT (Ct) method and expressed as $2^{-\Delta\Delta Ct}$ with the control group arbitrarily set as 1. Primer sequence is listed in Supplementary Table 1.

**Statistics**. Results were expressed as mean ± S.E.M or mean ± SD as noted. Bartlett's test was used to determine equal variance. Two-way ANOVA and Tukey post hoc test or Student's $t$-test was used to calculate the $p$-value as noted in each figure legend. A $p < 0.05$ was considered statistically significant.

**Reporting summary**. Further information on research design is available in the Nature Research Reporting Summary linked to this article.

## Data availability

The authors declare that data supporting the findings of this study are available within the paper and its Supplementary Information files. Extended information is available either in the Source Data file or upon reasonable request. Available source data underlying all figures except metabolomics data Fig. 7e–h are provided as a Source Data file. Metabolomics data Fig. 7e–h will be available upon reasonable request.

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

## Acknowledgements

This study is supported in part by NIH grant 1R01 DK117965-01A1 (T.L.), R01 AA020518-08 (W.-X.D.), and National Center for Research Resources Grant 5P20RR021940-07 and National Institute of General Medical Sciences Grant 8 P20 GM103549-07.

## Author contributions

Y.W., F.L., D.J.M., C.C., X.C., T.J., Y.Z., and H.-M.N. designed and performed the experiments. S.G. performed bioinformatics and statistical analysis of the metabolomics data. M.C. provided human liver samples and helped with manuscript writing. H.-M.N. and W.-X.D. provided key reagents, supervised the study, and reviewed the manuscript. T.L. supervised the study and wrote the manuscript.

## Competing interests

The authors declare no competing interests.
