## [Peer Review File · Nature Communications]

Peer Review File - Reviewers' comments first round:

Reviewer #1 (Remarks to the Author):

This study from Wang et al. reports that TFEB, a recently identified key regulator of lysosomal biogenesis and autophagy, is a novel inducer for hepatic bile acid synthesis downstream of FGF19/FGFR4 pathway. This study is very novel and significant because this is a new link to link bile acid synthesis with lysosomal stress.

The authors showed that in mice and primary human hepatocytes, TFEB strongly enhanced Cyp7a1/CYP7A1 gene expression, leading to a significantly increased size of bile acid pool, which helps mice resistant to diet-induced hepatic cholesterol accumulation. EMSA and ChIP assays were then used to show that TFEB directly binds to CYP7A1 promoter regions in human and mouse livers. Moreover, the study showed that TFEB nuclear translocation was feed-forwardly stimulated by acute cholesterol loading, presumably as a result of lysosomal impairment, but was inhibited by FGF19 signaling via activating of mTOR and ERK. This study provides evidence identifying TFEB as a new stimulating factor in regulating bile acid synthesis.

The significance of TFEB in liver health is implicated with data showing that TFEB activation was impaired in livers of mice and humans with NASH. Furthermore, treatment of mice with an ASBT inhibitor, which disrupted the FGF15-mediated repressive effect of TFEB, stimulated hepatic TFEB nuclear translocation and prevented hepatic steatosis.

This is a well conducted study with mechanistic approaches by using gain- and loss-of-function mouse models, as well as human studies in primary human hepatocytes and normal and NAFLD livers. The identification of TFEB as a new inducer of bile acid synthesis is a highly significant because novel knowledge is urgently needed to understand bile acid homeostasis and gut-liver crosstalk. In addition, the finding that an intestine ASBT inhibitor could modulate this novel signaling axis to induce hepatic TFEB provides a new pathway linking the gut-liver bile acid pathway to several aspects of liver metabolism, which is also of translational significance given the potential clinical applications of ASBT inhibitor in diabetes, fatty liver disease and cholestasis.

This study would be further strengthened once these questions/comments could be addressed:

- It is interesting that acute cholesterol treatment caused rapid TFEB nuclear translocation. The authors showed in their previous publication and here that cholesterol impaired lysosomal function. Since the results also showed that mTOR and ERK were key regulators of TFEB nuclear and cytosolic distribution and mediated FGF19 regulation of TFEB, the effects of cholesterol loading on these signaling pathways need to be investigated, especially we know that the cholesterol homeostasis is different in mice and humans.
- The authors showed nicely that TFEB was a strong inducer of CYP7A1 and bile acid synthesis. However, bile acids and FGF19 have been reported to inhibit CYP7A1 via various redundant pathways involving hepatic and intestine-initiated signaling and epigenetic chromatin modifications. Therefore, additional experiments need be done to evaluate if TFEB is required or dispensable for bile acids and FGF19 to inhibit CYP7A1 in hepatocytes via TFEB overexpression or knockdown approaches in hepatocytes or HepG2 cells they used in the study. It will be also helpful to identify the activities of TFEB in mice with modulations of FGF15 levels.
- TFEB overexpression altered bile acid composition without changing CYP8B1 expression. Any explanation?
- Fig 6D. Is there an increased cytosolic TFEB? Please provide densitometry analysis.

Reviewer #2 (Remarks to the Author):

In this paper Wang et al. show that the transcription factor TFEB promotes bile acid biogenesis through the regulation of CYP7A1 gene expression. They also show that modulation of TFEB in mice significantly influences liver cholesterol homeostasis and bile acid production/composition. Hepatocytes loaded with cholesterol induce TFEB nuclear translocation, while FGF19 shows opposite effects. In vivo, pharmacological inhibition of bile acid uptake induces TFEB nuclear translocation and improves hepatic metabolic homeostasis in western diet-fed mice. Overall, this is an interesting study that identifies a previously unrecognized role for TFEB in bile acid synthesis and cholesterol metabolism. The paper is clearly written and the data are generally of good quality. However, the following points should be addressed to better demonstrate the physiological involvement of TFEB in bile acid homeostasis:

1) The authors claim that cholesterol induces TFEB nuclear translocation. However, based on the data presented, this may be a consequence of an "artificial" induction of lysosomal stress due to excessive cholesterol administration in vitro, rather than a physiological response to cholesterol in hepatocytes. The authors should support their conclusion by analyzing TFEB localization in a more physiological context (i.e. in vivo in the liver of the hypercholesterolemia mouse model).

2) Similarly, there is no demonstration that FGF15 physiologically regulates TFEB. The observation that stimulation of hepatocytes with FGF19/15 promotes TFEB cytosolic retention should also be corroborated in vivo showing that the administration of FGF15/19 in the hypercholesterolemia mouse model promotes TFEB cytosolic retention.

3) The mechanism of TFEB nuclear translocation in cholesterol-treated cells is unclear. The authors should investigate whether cholesterol overload affects the known signaling pathways controlling TFEB nuclear translocation (i.e. mTORC1 activity and localization). Also fig. 5 contains several puzzling data: in B, FGF19 does not induce TFEB cytosolic re-localization in U0126-treated samples, while opposite results are shown in panel E. Based on the results shown in panel H it is impossible to determine the relative importance of S211 and S142 for TFEB nuclear translocation. To make this comparison the authors should perform TFEB localization studies by immunofluorescence in cholesterol-treated cells carrying S142A and S211A mutations and examine the data through a robust statistical analysis. In addition, subcellular fractionation data should be carefully quantified from multiple independent experiments. It will be important to compare the different effects of the TFEB mutants by loading all the nuclear extracts on the very same gel.

4) To claim that GSK treatment is indeed working through TFEB activation the authors should repeat some of the key experiments of figure 7, including expression of TFEB target genes, by silencing TFEB in mice treated with GSK, to check whether the response to GSK is blunted.

Reviewer #3 (Remarks to the Author):

In the current work by Wang et al, the authors illustrated a TFEB-mediated gut-liver signaling axis that regulates cholesterol and bile acid homeostasis. It was found that TFEB induces CYP7A1, increases the conversion of cholesterol to bile acids, and prevents hepatic cholesterol accumulation and hypercholesterolemia. In addition, the authors demonstrated that cholesterol-induced lysosomal stress feed-forward activates TFEB via promoting TFEB nuclear translocation. The authors found that bile acid-induced FGF19/15 feedback inhibits TFEB nuclear translocation. Furthermore, the authors showed that an ASBT inhibitor decreases bile acid uptake and decreases ileal FGF15 production, but enhances hepatic TFEB function and improves hepatic metabolic homeostasis. All results support the conclusion.

Comments and suggestions:

1. Suggest the deletion of "and may be pharmacologically modulated by targeting the intestinal bile acid transport" from the last sentence of abstract. This part can be highlighted in the discussion.
2. Dose justification is needed for GSK (2 mg/kg).
3. More details of statistics are needed for Fig legends. Different levels of p-values should be

provided to show the statistical significance.

4. A higher magnification is needed to show hepatic lipid accumulation in Fig 7B? Or oil-red-O staining can be used.

5. CYP7A1 should be included in summary Fig 8 by replacing "bile acid synthesis".

6. More discussions are suggested to address the clinical relevance of the findings in this work.

7. Add "Supplemental" to the legend of Fig 9 (page 42). In addition, more information should be provided in this Fig. For example, what metabolites contribute to the group separation? Any bile acids or cholesterol (TC, FC, or CE) among the top ranking?

We thank all three reviewers for their constructive comments. We have conducted additional in vitro and in vivo experiments to address the raised concerns. Below, we first summarized the major changes in the revised manuscript and provided a point-to-point response to major review critics. We hope the reviewers find the major concerns adequately addressed and the revised manuscript has been improved.

Major changes:

1. We conducted in vivo experiments to show that FGF19 administration activated mTOR and ERK signaling and decreased TFEB nuclear translocation in mouse livers in vivo. **(Added new Fig 5A, 5B, 5C, 5D)**
2. We have evaluated the relative role of TFEB in mediating FGF19 repression of CYP7A1 in control and hepatic TFEB deficient mice treated with vehicle and FGF19. **(Added new Fig 5E, 5F)**
3. We have used commercial p-TFEB S211 and p-TFEB S142 antibodies and demonstrated that FGF19 treatment resulted in increased TFEB phosphorylation at S211 and S142 and that FGF19-activated mTOR and ERK signaling plays a key role in mediating TFEB phosphorylation. **(Added new Fig 4D, 4E, 4F and Supplemental Fig 3C and 3D)**
4. We have used suggested imaging and sub-cellular fractionation approach in combination with mutagenesis and demonstrated the relative role of S211 and S142 in regulating TFEB cytosolic and nuclear localization. **(Added new Fig 4G, 4H, 4I)**
5. We have used hepatic TFEB deficient mice and controls challenged with Western diet and evaluated the relative role of hepatic TFEB activation in GSK regulation of cholesterol homeostasis. **(Added new Fig 9F, 9G, 9H)**
6. We have analyzed hepatic TFEB nuclear localization in high cholesterol/fat Western diet fed mice and found that under this pathophysiologically relevant NAFLD condition adaptive hepatic TFEB nuclear translocation was observed in more advanced but not early stage of diet-induced NAFLD. **(Added new Supplemental Fig 1F, 1G, 1H, 1I)**
7. We have determined the effects of cholesterol treatment on mTOR and ERK signaling. **(Added new Supplemental Fig 1D, 1E)**
8. We have rearranged the Fig sequence to incorporate new data and have added new discussion in the text. All changes are in red font.

Point-to-point response

Reviewer #1.

This study from Wang et al. reports that TFEB, a recently identified key regulator of lysosomal biogenesis and autophagy, is a novel inducer for hepatic bile acid synthesis downstream of FGF19/FGFR4 pathway. This study is very novel and significant because this is a new link to link bile acid synthesis with lysosomal stress.

The authors showed that in mice and primary human hepatocytes, TFEB strongly enhanced Cyp7a1/CYP7A1 gene expression, leading to a significantly increased size of bile acid pool, which helps mice resistant to diet-induced hepatic cholesterol accumulation. EMSA and ChIP assays were then used to show that TFEB directly binds to CYP7A1 promoter regions in human and mouse livers. Moreover, the study showed that TFEB nuclear translocation was feed-forwardly stimulated by acute cholesterol loading, presumably as a result of lysosomal impairment, but was inhibited by FGF19

signaling via activating of mTOR and ERK. This study provides evidence identifying TFEB as a new stimulating factor in regulating bile acid synthesis.

The significance of TFEB in liver health is implicated with data showing that TFEB activation was impaired in livers of mice and humans with NASH. Furthermore, treatment of mice with an ASBT inhibitor, which disrupted the FGF15-mediated repressive effect of TFEB, stimulated hepatic TFEB nuclear translocation and prevented hepatic steatosis.

This is a well conducted study with mechanistic approaches by using gain- and loss-of-function mouse models, as well as human studies in primary human hepatocytes and normal and NAFLD livers. The identification of TFEB as a new inducer of bile acid synthesis is a highly significant because novel knowledge is urgently needed to understand bile acid homeostasis and gut-liver crosstalk. In addition, the finding that an intestine ASBT inhibitor could modulate this novel signaling axis to induce hepatic TFEB provides a new pathway linking the gut-liver bile acid pathway to several aspects of liver metabolism, which is also of translational significance given the potential clinical applications of ASBT inhibitor in diabetes, fatty liver disease and cholestasis.

This study would be further strengthened once these questions/comments could be addressed:

- 1. It is interesting that acute cholesterol treatment caused rapid TFEB nuclear translocation. The authors showed in their previous publication and here that cholesterol impaired lysosomal function. Since the results also showed that mTOR and ERK were key regulators of TFEB nuclear and cytosolic distribution and mediated FGF19 regulation of TFEB, the effects of cholesterol loading on these signaling pathways need to be investigated, especially we know that the cholesterol homeostasis is different in mice and humans.**

Response: We have shown in new Supplemental 1D-E that cholesterol treatment in time course did not affect mTOR and ERK signaling in HepG2 cells. In addition, we showed in new Fig 4D-F that cholesterol treatment did not affect p-TFEB at S142 and S211. Since under this condition cholesterol induced strong nuclear TFEB translocation, it suggests that cholesterol loading-induced TFEB nuclear translocation is independent of mTOR and ERK signaling, which is in contrast to FGF19 regulation of TFEB nuclear localization. (Please also see our response to reviewer #2 question 3A.)

- 2. A. The authors showed nicely that TFEB was a strong inducer of CYP7A1 and bile acid synthesis. However, bile acids and FGF19 have been reported to inhibit CYP7A1 via various redundant pathways involving hepatic and intestine-initiated signaling and epigenetic chromatin modifications. Therefore, additional experiments need be done to evaluate if TFEB is required or dispensable for bile acids and FGF19 to inhibit CYP7A1 in hepatocytes via TFEB overexpression or knockdown approaches in hepatocytes or HepG2 cells they used in the study. B. It will be also helpful to identify the activities of TFEB in mice with modulations of FGF15 levels.**

Response: A. We have shown in new Fig 5E-F that knockdown of hepatic TFEB reduced CYP7A1 expression but did not affect the maximal FGF19 inhibition of CYP7A1 expression. This result is consistent with previous studies that redundant pathways are

involved in FGF19 inhibition of CYP7A1 and TFEB is partially mediating FGF19 inhibition of CYP7A1. **B.** We have shown in new Fig 5A-D that FGF19 administration activated mTOR and ERK, repressed CYP7A1 and inhibited nuclear TFEB localization in mice in vivo, which thus supported our in vitro findings that FGF19 inhibited TFEB nuclear localization in hepatocytes.

3. TFEB overexpression altered bile acid composition without changing CYP8B1 expression. Any explanation?

Response: We showed that hepatic CYP8B1 expression was not altered, suggesting that altered bile acid composition may not be a result of altered cholic acid synthesis in the liver. Since TFEB strongly increased bile acid pool size by ~2-fold (Fig 8D), we speculate that altered bile acid composition could be secondary to increased amount of bile acids circulating in the enterohepatic circulation. Bile acids are known to modulate gut microbiome via their anti-microbial activity and FXR signaling. Supplemental Fig 10 A-B showed increased unconjugated CA and T-DCA. Conversion of T-CA to T-DCA involves bacterial enzyme-mediated T-CA deconjugation and subsequent CA dehydroxylation. In addition, decreased T-MCA and increased MCA also suggest increased primary bile acid deconjugation. Therefore, we think it may be worthwhile to further investigate the gut microbiome changes in TFEB overexpressing mice. However, this requires substantial additional work that is out of the current scope of the study. We have discussed the changes of bile acid composition in the text but did not extensively discuss the possible underlying causes because of the lack of supporting experimental evidence. We are currently developing experimental approaches to study microbiome in mouse models and will follow up on these findings in future investigations.

4. Fig 6D. Is there an increased cytosolic TFEB? Please provide densitometry analysis

Response: We have added densitometry graph in Fig 6D. We showed that nuclear TFEB protein increased by ~4-fold, which was not a result of increased total TFEB protein in mouse livers.

Reviewer #2

In this paper Wang et al. show that the transcription factor TFEB promotes bile acid biogenesis through the regulation of CYP7A1 gene expression. They also show that modulation of TFEB in mice significantly influences liver cholesterol homeostasis and bile acid production/composition. Hepatocytes loaded with cholesterol induce TFEB nuclear translocation, while FGF19 shows opposite effects. In vivo, pharmacological inhibition of bile acid uptake induces TFEB nuclear translocation and improves hepatic metabolic homeostasis in western diet-fed mice.

Overall, this is an interesting study that identifies a previously unrecognized role for TFEB in bile acid synthesis and cholesterol metabolism. The paper is clearly written and the data are generally of good quality. However, the following points should be addressed to better demonstrate the physiological involvement of TFEB in bile acid homeostasis:

- 1. The authors claim that cholesterol induces TFEB nuclear translocation. However, based on the data presented, this may be a consequence of an “artificial” induction of lysosomal stress due to excessive cholesterol administration in vitro, rather than a physiological response to cholesterol in hepatocytes. The authors should support their conclusion by analyzing TFEB localization in a more physiological context (i.e. in vivo in the liver of the hypercholesterolemia mouse model).**

Response: As suggested, we have analyzed nuclear and cytosolic TFEB in livers of 8 week and 16 week high cholesterol/fat Western diet -fed mice and results are presented in Supplemental Fig 1F-I. We found that 16-week Western diet feeding, but not 8-week Western diet feeding, resulted in increased nuclear TFEB and decreased cytosolic TFEB, supporting adaptive TFEB nuclear translocation. These results suggest that adaptive TFEB activation occurred under more chronic patho-physiologically relevant NAFLD condition in vivo. We think that 8-week Western diet fed mice did not show increased nuclear TFEB likely because it represented early stage mild simple steatosis without appreciable hepatocellular stress and injury and thus lack adaptive TFEB nuclear translocation. This is not unexpected because liver has a very high capacity to maintain hepatic cholesterol homeostasis and limit cholesterol-induced cellular stress by coordinating cholesterol esterification, cholesterol synthesis, efflux to circulation, biliary secretion and bile acid synthesis. However, these protective mechanisms may be overwhelmed upon prolonged high cholesterol/fat diet challenge which is likely the case after 16-week WD feeding. Our findings are in agreement with the reviewer’s comment that cholesterol treatment in hepatocytes in vitro is a severe form of intracellular cholesterol accumulation that is associated with lysosomal stress and cell injury and this in vitro model does not represent simple steatosis in the early stage of NAFLD. We also cited references #42, 50, 51 to support that lysosomal stress was associated with advanced human NASH and cholesterol has been suggested to play a causative role in liver injury and inflammation. We acknowledge that under advanced NAFLD/NASH condition the causes of lysosomal stress are likely complex and multi-factorial. However, findings from cholesterol-laden hepatocyte models suggest that advanced hepatic cholesterol accumulation likely plays an important role in the TFEB adaptive response in NAFLD. Corresponding discussions are added in text in red font.

- 2. Similarly, there is no demonstration that FGF15 physiologically regulates TFEB. The observation that stimulation of hepatocytes with FGF19/15 promotes TFEB cytosolic retention should also be corroborated in vivo showing that the administration of FGF15/19 in the hypercholesterolemia mouse model promotes TFEB cytosolic retention. (Same question is raised by reviewer #1)**

Response: We have administered FGF19 to mice for 6 hours and demonstrated that FGF19 activated hepatic mTOR and ERK, inhibited CYP7A1, and resulted in decreased nuclear TFEB and increased cytosolic TFEB. These results suggest that FGF19 regulates hepatic TFEB nuclear localization in vivo. These data are shown in new Fig 5A-D.

- 3. The mechanism of TFEB nuclear translocation in cholesterol-treated cells is unclear. The authors should investigate whether cholesterol overload affects the known signaling pathways controlling TFEB nuclear translocation (i.e. mTORC1 activity and localization). Also Fig. 5 contains several puzzling data: in B, FGF19 does not induce TFEB cytosolic re-localization in U0126-treated samples, while**

opposite results are shown in panel E. Based on the results shown in panel H it is impossible to determine the relative importance of S211 and S142 for TFEB nuclear translocation. To make this comparison the authors should perform TFEB localization studies by immunofluorescence in cholesterol-treated cells carrying S142A and S211A mutations and examine the data through a robust statistical analysis. In addition, subcellular fractionation data should be carefully quantified from multiple independent experiments. It will be important to compare the different effects of the TFEB mutants by loading all the nuclear extracts on the very same gel.

Response:

A. In our continued study, we have identified commercial antibodies that detected p-TFEB S142 and p-TFEB S211. With these antibodies, we demonstrated in the revised manuscript new Fig 4D, 4E and 4F that FGF19 indeed induced TFEB S142 and S211 phosphorylation and that inhibition of mTOR or ERK decreased S211 and S142 phosphorylation. These new data serve as key evidence that FGF19 phosphorylates TFEB to cause TFEB cytosolic retention. Unfortunately, we did not detect any bands corresponding to P-TFEB using mouse liver tissues from mice injected with vehicle or FGF19. The manufacturers also claimed that these two phospho-TFEB antibodies only reacted with human TFEB.

B. We have shown in the new Supplemental 1D-E that cholesterol treatment in time course did not affect mTOR and ERK signaling. Consistently, we have shown in new Fig 4D-F that cholesterol treatment did not affect p-TFEB at S142 and S211. Since under this condition cholesterol induced strong nuclear TFEB translocation, it suggested that cholesterol loading-induced TFEB nuclear translocation was independent of mTOR and ERK signaling and S142/S211 phosphorylation, which is in contrast to FGF19 regulation of TFEB nuclear localization. We speculate that in contrast to nutrient signaling (mTOR/ERK) regulation of TFEB, cholesterol induced lysosomal stress could regulate TFEB via crosstalk with novel signaling cascade and new TFEB phosphorylation sites, and possibly signaling/phosphorylation-independent mechanisms. Intracellular cholesterol trafficking is a highly dynamic and complex process with limited investigative tools. We hope that the reviewer agrees that further investigation along this line of research will require significant amount of additional work and may be out of the scope of the current report. The mechanism of cholesterol regulation of TFEB function is an important question remains to be elucidated and we plan to continue this line of research in future investigations.

C. As suggested, we have used immunofluorescence imaging approach and sub-cellular fractionation as suggested to determine the relative involvement of S142 and S211 in regulating TFEB nuclear localization downstream of FGF19 (New Fig 4G-I). In addition, the differential phosphorylation of S211 and S142 by ERK and mTOR (new Fig 4D-F, new Supplemental Fig 3C-D) also help shed new insight on the relative importance of S211 and S142 in TFEB nuclear localization. A few key findings from these new data include: 1. Both S211A and S142A showed increased basal nuclear localization than WT TFEB, suggesting that both are involved in regulating TFEB cytosolic retention; 2. Mutagenesis abolishing S211 phosphorylation increased TFEB nuclear abundance from baseline ~30-35% to ~70% TFEB while abolishing S142 phosphorylation increased nuclear TFEB to only ~45%, supporting that S211 phosphorylation plays a key role in TFEB cytosolic retention; 3. Signaling inhibitor study showed that S211 phosphorylation is targeted by both mTOR and ERK and is required for FGF19 inhibition of TFEB nuclear localization; 4. S142 is targeted by ERK but not mTOR. Blocking mTOR signaling caused 70% TFEB nuclear localization without decreasing S142 phosphorylation,

suggesting S142 phosphorylation plays a relatively minor role than S211 phosphorylation in TFEB cytosolic retention.

D. The reviewer pointed out that ERK inhibitor prevented FGF19 inhibition of nuclear TFEB localization in HepG2 cells but not in human hepatocytes despite that ERK inhibitor increased TFEB nuclear translocation in both cell types. We agree with the reviewer that there is some cell type-specific response to ERK inhibition and FGF19 treatment in HepG2 cells and primary human hepatocytes, the cause of which we still could not clearly explain. FGF19 is known to activate many cellular signaling pathways. It is reasonable to speculate that FGF19 could activate other ERK independent signaling mechanisms to decrease nuclear TFEB in the presence of U0126 in primary human hepatocytes. The lack of similar response in HepG2 cells may be due to fundamental differences in cell signaling crosstalk in HepG2 cells that remains to be investigated. On the other hand, despite some cell-type dependent discrepancy, TFEB responses to cholesterol, signaling inhibitors and FGF19 have been very consistent in these two cell types, and including two cell types in our study improved the robustness of our findings. Further, our new in vivo data showed that FGF19 administration strongly inhibited nuclear TFEB in chow-fed mice, suggesting that FGF19 regulates TFEB nuclear translocation under chow-fed physiological condition. Therefore, we think that data from several lines of experiments generally support our conclusion.

- 4. To claim that GSK treatment is indeed working through TFEB activation the authors should repeat some of the key experiments of Figure 7, including expression of TFEB target genes, by silencing TFEB in mice treated with GSK, to check whether the response to GSK is blunted.**

Response: In the original submission, we have shown (Fig 9A-E in revised manuscript) that in response to WD challenge hepatic TFEB deficiency promoted hepatic VLDL secretion and significantly sensitized mice to development of hypercholesterolemia. Along this line of findings, we now show in new Fig 9F-H that preventing GSK induction of hepatic TFEB by silencing hepatic TFEB completely prevented GSK from lowering plasma cholesterol, rendering mice hypercholesterolemic in the presence of GSK treatment. In addition, we have shown in Fig 8 that increased hepatic TFEB significantly attenuated Western diet-induced cholesterol dysregulation independent of hepatic steatosis. These results collectively suggest that hepatic TFEB activation partially contributes to the beneficial effects of GSK in maintaining overall cholesterol homeostasis, although GSK is expected to act via other mechanisms to control metabolic homeostasis. We have added relevant discussion on how TFEB may help maintain overall cholesterol homeostasis by coordinately controlling lysosomal biogenesis and bile acid synthesis and how impairment of these mechanisms contributes to dysregulation of cholesterol homeostasis in the Discussion Section.

Reviewer #3.

In the current work by Wang et al, the authors illustrated a TFEB-mediated gut-liver signaling axis that regulates cholesterol and bile acid homeostasis. It was found that TFEB induces CYP7A1, increases the conversion of cholesterol to bile acids, and prevents hepatic cholesterol accumulation and hypercholesterolemia. In addition, the authors demonstrated that cholesterol-induced lysosomal stress feed-forward activates TFEB via promoting TFEB nuclear translocation. The authors found that bile acid-induced FGF19/15 feedback inhibits TFEB nuclear translocation. Furthermore, the

authors showed that an ASBT inhibitor decreases bile acid uptake and decreases ileal FGF15 production, but enhances hepatic TFEB function and improves hepatic metabolic homeostasis. All results support the conclusion.

1. **Suggest the deletion of “and may be pharmacologically modulated by targeting the intestinal bile acid transport” from the last sentence of abstract. This part can be highlighted in the discussion.**

Response: We have deleted this sentence and added discussion in the text.

2. **Dose justification is needed for GSK (2 mg/kg).**

Response: The original publication (Reference #12) on the discovery of GSK showed that GSK was most effective in blocking intestinal bile acid uptake at doses ranging from 1 mg/kg to 10 mg/kg. We have tested 2 mg/kg, 4 mg/kg and 10 mg/kg in our studies and found that 2 mg/kg is effective in blocking intestine bile acid uptake. We have stated that this dose is in line with the published literature in the Material and Method section.

3. **More details of statistics are needed for Fig legends. Different levels of p-values should be provided to show the statistical significance.**

Response: Where applicable, we have shown three levels of p values (0.05, 0.01 and 0.001) in Figures.

4. **A higher magnification is needed to show hepatic lipid accumulation in Fig 7B? Or oil-red-O staining can be used.**

Response: we added H&E with higher magnification to show differences in hepatic lipid droplet accumulation in vehicle and GSK treated mice.

5. **CYP7A1 should be included in summery Fig 8 by replacing “bile acid synthesis”.**

Response: We have revised this in new Fig 10.

6. **More discussions are suggested to address the clinical relevance of the findings in this work.**

Response: We have made significant revision in the Discussion Section to discuss the pathological and therapeutic relevance and also added additional discussion based on new data included to the revised manuscript. All changes are in red font.

7. **Add “Supplemental” to the legend of Fig 9 (page 42). In addition, more information should be provided in this Fig. For example, what metabolites contribute to the group separation? Any bile acids or cholesterol (TC, FC, or CE) among the top ranking?**

Response: The original Supplemental Fig 9 becomes new Supplemental Fig 7A. We have included detailed pathway analysis in new Supplemental Fig 7B-D to show top altered pathways in each cluster which included lipid metabolism pathways. Changes of specific lipotoxic lipids were also shown in Fig 7.

REVIEWERS' COMMENTS second round:

Reviewer #1 (Remarks to the Author):

The authors have made substantial revision and addressed the previously raise concerns by additional in vitro and in vivo experiments.

Reviewer #2 (Remarks to the Author):

The authors have satisfactorily responded to points 1, 2 and 4. However, I have serious concerns on the data addressing point 3. In particular, the data in figure 4 have several problems, which may be due to the use of unreliable reagents to monitor phosphorylation of endogenous TFEB. It is very well established (over 50 publications) that both S142 and S211 are phosphorylated by mTORC1, and that S142 but not S211 is also phosphorylated by ERK (reviewed in Puertollano et al. 2018 EMBO J.). The data in figure 4D-F are showing opposite results that, in my opinion, are due to poor specificity of the phospho-antibodies when used to monitor phosphorylation of endogenous TFEB. In particular Figure 4D shows no effect of Torin1 on the phosphorylation of S142. This result is in contrast with what was previously reported by several groups. In addition, no TFEB molecular weight shift was detectable in the presence of cholesterol, FGF19, Torin1 or U0126. Furthermore, the differences observed between several cell lines indicate that there is no a clear regulation of TFEB phosphorylation by cholesterol and FGF19 treatment. This is not surprising given that a growing list of kinases and phosphorylation events are known to occur on TFEB. Given that the authors have provided several in vivo data demonstrating the importance of cholesterol and FGF19 for TFEB activation, and that FGF19 induce both ERK and mTORC1 signaling, I suggest to remove the data in Fig. 4D-F from the paper, thus avoiding to go into the details of phosphorylation of specific serines. Alternatively, the authors should repeat the experiments in fig 4D-F using cells (both HepG2 and primary human hepatocytes) stably infected with TFEB construct, and verify the specificity of the antibodies using Serine to Alanine mutant proteins.

Reviewer #3 (Remarks to the Author):

None

Response to review comments

Reviewer #1 (Remarks to the Author):

The authors have made substantial revision and addressed the previously raise concerns by additional in vitro and in vivo experiments.

Reviewer #2 (Remarks to the Author):

The authors have satisfactorily responded to points 1, 2 and 4. However, I have serious concerns on the data addressing point 3. In particular, the data in figure 4 have several problems, which may be due to the use of unreliable reagents to monitor phosphorylation of endogenous TFEB. It is very well established (over 50 publications) that both S142 and S211 are phosphorylated by mTORC1, and that S142 but not S211 is also phosphorylated by ERK (reviewed in Puertollano et al. 2018 EMBO J.). The data in figure 4D-F are showing opposite results that, in my opinion, are due to poor specificity of the phospho-antibodies when used to monitor phosphorylation of endogenous TFEB. In particular Figure 4D shows no effect of Torin1 on the phosphorylation of S142. This result is in contrast with what was previously reported by several groups. In addition, no TFEB molecular weight shift was detectable in the presence of cholesterol, FGF19, Torin1 or U0126. Furthermore, the differences observed between several cell lines indicate that there is no a clear regulation of TFEB phosphorylation by cholesterol and FGF19 treatment. This is not surprising given that a growing list of kinases and phosphorylation events are known to occur on TFEB. Given that the authors have provided several in vivo data demonstrating the importance of cholesterol and FGF19 for TFEB activation, and that FGF19 induce both ERK and mTORC1 signaling, I suggest to remove the data in Fig. 4D-F from the paper, thus avoiding to go into the details of phosphorylation of specific serines. Alternatively, the authors should repeat the experiments in fig 4D-F using cells (both HepG2 and primary human hepatocytes) stably infected with TFEB construct, and verify the specificity of the antibodies using Serine to Alanine mutant proteins.

Response: Following the reviewer's suggestion, we have removed Fig 4d-f from the manuscript. We have made corresponding changes in the text to reflect these changes. We will investigate the mechanisms by which cholesterol and FGF19 regulate TFEB function in future follow-up studies.

Reviewer #3 (Remarks to the Author):

None